# Choose Your Model Size: Any Compression of Large Language Models Without Re-Computation

**Martin Genzel**[*]                                                            *martin.genzel@merantix-momentum.com*
*Merantix Momentum, Berlin, Germany*

**Patrick Putzky**[*]                                                          *patrick.putzky@merantix-momentum.com*
*Merantix Momentum, Berlin, Germany*

**Pengfei Zhao**[*,†]                                                                    *pzhao@atb-potsdam.de*
*Understandable Machine Intelligence Lab*
*Leibniz Institute for Agriculture and Bioeconomy, Potsdam, Germany*

**Sebastian Schulze**                                                       *sebastian.schulze@merantix-momentum.com*
*Merantix Momentum, Berlin, Germany*

**Mattes Mollenhauer**                                                    *mattes.mollenhauer@merantix-momentum.com*
*Merantix Momentum, Berlin, Germany*

**Robert Seidel**[†]

**Stefan Dietzel**                                                            *stefan.dietzel@merantix-momentum.com*
*Merantix Momentum, Berlin, Germany*

**Thomas Wollmann**                                                        *thomas.wollmann@merantix-momentum.com*
*Merantix Momentum, Berlin, Germany*

[*] *Equal Contribution*
[†] *Work done while at Merantix Momentum*

**Reviewed on OpenReview:** *https://openreview.net/forum?id=Y6hdYf8tsg*

## Abstract

The adoption of Foundation Models in resource-constrained environments remains challenging due to their large size and inference costs. A promising way to overcome these limitations is post-training compression, which aims to balance reduced model size against performance degradation. This work presents *Any Compression via Iterative Pruning* (`ACIP`), a novel algorithmic approach to determine a compression-performance trade-off from a single stochastic gradient descent run. To achieve parameter efficiency, we use an SVD-reparametrization of linear layers and iteratively prune their singular values with a sparsity-inducing penalty. Importantly, the pruning order of the parameters is used to derive a global score map that allows compressing a model to any target size without re-computation. We evaluate `ACIP` on a large selection of open-weight LLMs and downstream tasks, demonstrating state-of-the-art results compared to existing factorization-based compression methods. We also show that `ACIP` seamlessly complements common quantization-based compression techniques.

## 1 Introduction

Post-training compression of Foundation Models, especially Large Language Models (LLMs), promises access to powerful tools where resources are limited, e.g., in automotive systems, mobile deployments, or on shop

floors (Gholami et al., 2022). Typical reasons for resource scarcity include constrained access to hardware, monetary limitations, high inference speed requirements, and environmental concerns (Hohman et al., 2024).

The original promise of model compression was to eliminate redundant parameters, resulting in almost lossless methods (Han et al., 2016). While working well for models trained on smaller datasets, this hypothesis does not hold up anymore in the era of LLMs and scaling laws (Allen-Zhu & Li, 2024). For modern "densely trained" models, compression is almost always lossy, leading to a fundamental trade-off between model size and downstream performance. While characterizing this trade-off supports practitioners in deployment decisions (Boggust et al., 2025), the scientific literature typically focuses on benchmarks at preset compression levels (Zhu et al., 2024). This gap between research and practice implies that, for model users, the process is often perceived as a "black box", requiring significant expertise and trial-and-error to identify an acceptable setup. We argue for the opposite approach, one that empowers users to seamlessly customize a compression algorithm for their specific use cases.

**Any Compression**

With this motivation in mind, we advocate for methods that permit *Any Compression* of pre-trained models. Here, '*Any*' signifies an algorithm's ability to scale a given base model to an arbitrary target size, guided by the user's specific needs and limitations, rather than the algorithm dictating possible sizes. To facilitate decision-making, such an algorithm must efficiently reveal the compression-performance trade-off without extensive re-computation. In existing post-training compression approaches, we identify two practical challenges to achieving Any Compression.

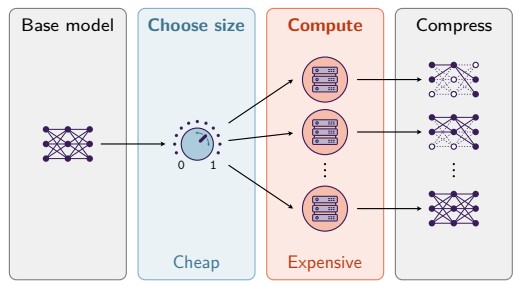

(a) Conventional Model Compression

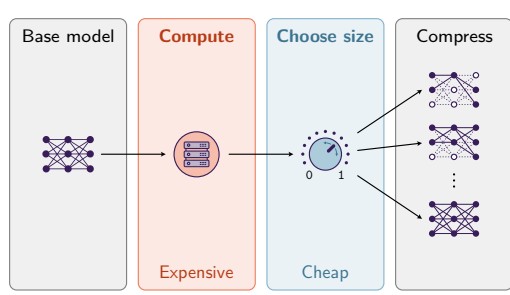

(b) Any Compression

**Problem 1** (Preset compression rates)**.** The size reduction of large Foundation Models is a prominent research area, with existing methods largely falling into three categories: *knowledge distillation* (training a smaller student model), *quantization* (reducing numerical precision), and *parameter pruning* (removing redundant weights) — for a comprehensive survey, we refer to Zhu et al. (2024). While effective, these approaches often impose constraints that conflict with the goal of achieving Any Compression as they are restricted to preset (discrete) compression factors. Indeed, knowledge distillation is limited to a single compression rate defined by the fixed size of the student model (Hinton et al., 2015). Quantization is limited by fixed bit-length reductions (typically from 16-bit to 4-bit or 8-bit representations) (Gholami et al., 2022; Dettmers

Figure 1: Compared to conventional compression algorithms (a), an Any Compression algorithm (b) swaps the computational calibration step and the decision step, so that models of different target sizes can be materialized without re-computation.

et al., 2023). Similarly, (unstructured) parameter pruning often relies on rigid sparsity patterns (e.g., $n{:}m$ sparsity) to ensure efficient memory allocation and hardware acceleration, thereby restricting the possible model sizes (Choquette et al., 2021). In practice, preset compression rates are especially problematic when deploying under a specific, non-discrete constraint, e.g., a fixed memory budget, forcing users into a costly "guess-and-check" cycle.

*Solution* (Structured pruning by weight factorization)**.** Weight factorization is a technique where (linear) model weights are decomposed into several sub-matrices (Eckart & Young, 1936). This approach enables fine-grained model compression and high parameter efficiency by pruning the inner dimensions of the resulting matrices (Yuan et al., 2024; Wang et al., 2024). A key advantage is its compatibility with standard hardware, since it only requires basic matrix-vector multiplications.

**Problem 2** (Different compression rates require re-computation)**.** Most existing post-training compression techniques are inherently inefficient for exploring the trade-off between model size and performance. Conven-

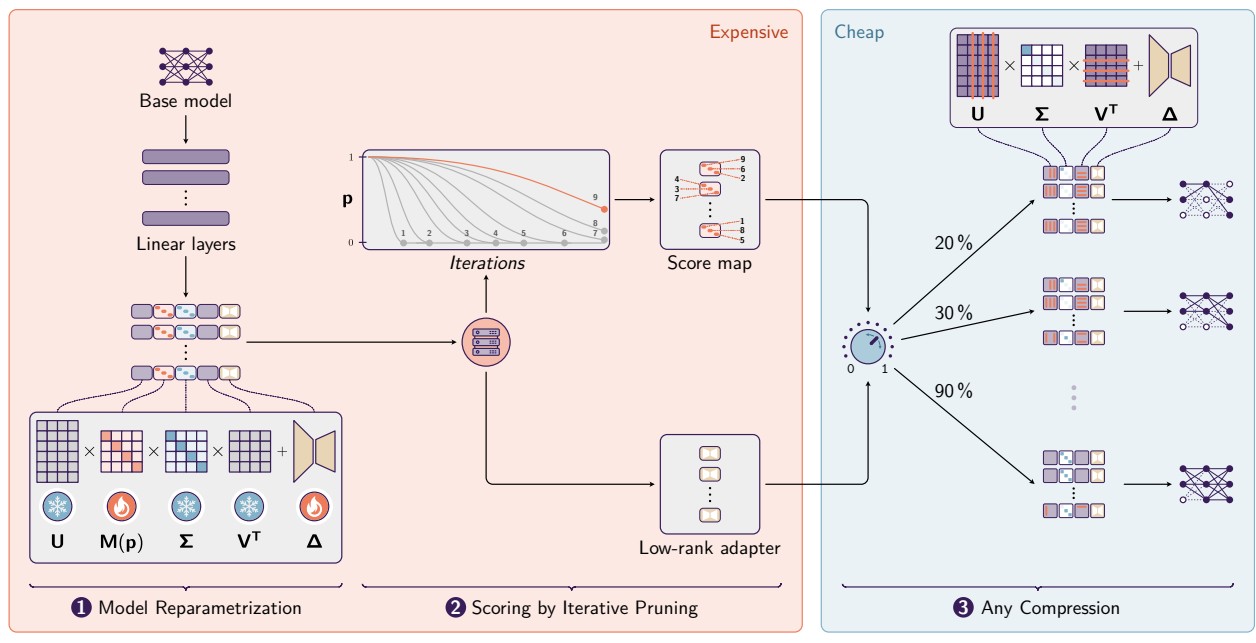

Figure 2: A visual overview of `ACIP`. ❶ The linear layers of the base model are reparametrized in terms of their singular value decomposition $\mathbf{U}\mathbf{M}\boldsymbol{\Sigma}\mathbf{V}^{\top}$, with a (binary) singular value mask $\mathbf{M} = \mathbf{M}(\mathbf{p})$ and a low-rank adapter $\boldsymbol{\Delta}$. ❷ An objective function is optimized via gradient descent over the mask parameters $\mathbf{p}$ and adapters $\boldsymbol{\Delta}$, where sparsity is induced on $\mathbf{p}$ by an increasing $\ell_1$-penalty. This leads to pruned entries in the mask $\mathbf{M}(\mathbf{p})$. The optimization path of $\mathbf{p}$ gives rise to a score map that determines the global importance of the singular values across the full model. Potential compression errors are compensated by $\boldsymbol{\Delta}$. ❸ Based on the parameter scores, the base model can be flexibly compressed to any target size by masking the entries of $\boldsymbol{\Sigma}$. The learned adapters $\boldsymbol{\Delta}$ are used as correction for any compression level.

tionally, a user selects a target compression rate and other hyperparameters, initiating a costly computational step that can take minutes to hours (Frantar et al., 2023; Sun et al., 2024; Yuan et al., 2024). Each new compression rate requires repeating this entire process, making a comprehensive evaluation of the trade-off landscape impractical (see Figure 1(a)).

*Solution* (Any Compression through amortization). We argue that a reversed workflow is preferable: a single, upfront computational investment that enables the subsequent materialization of a model at any compression rate in almost real-time (see Figure 1(b)). The initial effort, which could be handled by the model supplier, would empower users to efficiently select an optimal model instance for their needs at a negligible cost. The challenge, therefore, is to design algorithms that are compatible with this "compute once, compress dynamically" approach.

### Contributions and Overview

In this work, we introduce *Any Compression via Iterative Pruning* (`ACIP`),[1] which is specifically developed to address the above problems. To the best of our knowledge, `ACIP` is the first algorithm that enables large-scale model compression to any size in real-time without requiring re-computation or re-calibration.

To overcome Problem 1, `ACIP` follows a low-rank factorization strategy, based on pruning singular values of large linear layers. This particular form of structured parameter pruning allows for fine-grained compression levels and is parameter-efficient at the same time, as only singular values are altered. Compared to quantization and conventional (unstructured) pruning, an efficient implementation does not come with any restrictions to the underlying hardware, since it is purely based on standard matrix-vector multiplications.

---

[1]*Pronounced* like 'a sip' of coffee.

`ACIP` solves Problem 2 by explicitly decoupling an (optimization-based) pruning stage from the actual compression stage. The former can be viewed as a data-dependent calibration step, which estimates the global importance of all target parameters (the singular values of the base model layers) through an iterative pruning scheme. The resulting score map is then used to implement a simple compression step that enables the instantiation of models of any desired size without further computational costs. The detailed methodology and technicalities of `ACIP` are presented in Section 2 (see Figure 2 for a visual overview).

We empirically demonstrate the effectiveness of `ACIP` on a range of recent LLMs in Section 3, accompanied by a series of additional experiments in the supplementary material (Appendices C and D). In particular, we verify that our approach outperforms other factorization-based methods on multiple benchmarks (Section 3.2) and seamlessly complements quantization-based compression techniques (Section 3.4). In the spirit of scaling laws (Kaplan et al., 2020), `ACIP` provides consistent and robust size-performance trade-offs, which allows model users to predict downstream capabilities from a few data points.

Finally, we put our results in the broader context of related literature in Section 4 and conclude in Section 5 by outlining limitations as well as promising avenues for future research that build on our contributions.

## 2  Method

Figure 2 provides a schematic overview of *Any Compression via Iterative Pruning* (`ACIP`). In its initial stage, `ACIP` builds a reparametrization of large linear network layers by a singular value decomposition (SVD), which enables model compression through rank reduction (see Section 2.2.1 for details). Unlike existing SVD-based methods (Idelbayev & Carreira-Perpiñán, 2020; Yuan et al., 2024; Wang et al., 2024), Any Compression is achieved by decoupling the pruning and compression stages (cf. Figure 1(b)). More specifically, we construct a score map that establishes a global importance ranking of singular values across all linear layers within the network. The score map is derived by running a (stochastic) gradient descent on a sparsity-inducing objective, using the pruning order of the singular values as a proxy for feature importance (see Section 2.2.2). In an independent step, the score map can then be used to compress the base model to any desired size without re-computation (see Section 2.2.3).

### 2.1  Preliminaries

This section provides several preliminaries to better understand the algorithmic details of `ACIP`, which are presented in Section 2.2.

#### 2.1.1  Score-Based Parameter Pruning

As motivated above, the overarching goal of this work is to allow Any Compression of a pre-trained model by decoupling the computational stage from the compression stage. To this end, we use *score-based parameter pruning*, a framework that has been successfully applied to model compression since the 1980s (LeCun et al., 1989; Hassibi et al., 1993). In score-based pruning, a score map $\boldsymbol{\rho}$ is created that assigns an importance score $\rho_i$ to each target parameter $\theta_i$. Naturally, this approach gives rise to a (global) ranking of parameters that allows for model compression at any desired rate.

There are many ways to design useful score maps. For example, they can be derived based on curvature at a local optimum (LeCun et al., 1989; Hassibi et al., 1993) or by using hand-designed local features (Sun et al., 2024; Frantar & Alistarh, 2023). `ACIP` takes a novel, data-driven approach to score maps that does not require any handcrafting or feature engineering (see Section 2.2.2).

#### 2.1.2  Low-Rank Compression of Linear Layers

Linear layers are the molecular building blocks of modern machine learning models, typically accounting for more than 90% of model parameters in transformers (Vaswani et al., 2017), which makes them a natural target for compression. Accordingly, we consider linear layers of the form

$$\mathbf{y} = \mathbf{W}\mathbf{x} + \mathbf{b}, \tag{1}$$

where $\mathbf{W}$ is an $m \times n$ matrix, $\mathbf{b}$ is a bias term, and $\mathbf{x}$ and $\mathbf{y}$ are layer inputs and outputs, respectively. In this work, we specifically aim for a *low-rank compression* based on a matrix factorization such that

$$\mathbf{y} = \mathbf{P}\mathbf{Q}\mathbf{x} + \mathbf{b}, \tag{2}$$

where $\mathbf{P}$ and $\mathbf{Q}$ are matrices of sizes $m \times k$ and $k \times n$, respectively, and $k \ll \min(m, n)$. The layer parametrization in (2) may be interpreted as a (lossy) compression of the parametrization in (1), leading to a smaller memory footprint whenever $k(m + n) < mn$. Note that matrix factorization can be used to compress the parameters of any linear layer, including dense layers as well as efficiently parametrized layers such as convolutional layers (Idelbayev & Carreira-Perpiñán, 2020).

**Singular value decomposition.** To determine suitable low-rank factors $\mathbf{P}$ and $\mathbf{Q}$ in (2), we follow recent work on (large) model compression (Idelbayev & Carreira-Perpiñán, 2020; Yuan et al., 2024; Wang et al., 2024) and leverage a *singular value decomposition (SVD)*. SVD factorizes an $m \times n$ weight matrix $\mathbf{W}_l$ of rank $r \leq \min(m, n)$ at layer $l$ as

$$\mathbf{W}_l = \mathbf{U}_l \mathbf{\Sigma}_l \mathbf{V}_l^\top, \tag{3}$$

where $\mathbf{U}_l$ and $\mathbf{V}_l$ are $m \times r$ and $n \times r$ matrices of (orthonormal) singular vectors, and $\mathbf{\Sigma}_l$ is a $r \times r$ diagonal matrix containing the singular values $\mathbf{s}_l^{(i)} > 0$. Note that we consider the compact SVD here, which ignores the null-space vectors of the full SVD. We may reduce the inner dimension $r$ of the SVD to $k < r$ with

$$\mathbf{W}_l \approx \tilde{\mathbf{U}}_l \tilde{\mathbf{\Sigma}}_l \tilde{\mathbf{V}}_l^\top, \tag{4}$$

by defining $\tilde{\mathbf{U}}_l$ and $\tilde{\mathbf{V}}_l^\top$ to only contain $k$ singular vectors associated with a selected subset of singular values. Measuring the approximation error in terms of (Frobenius) matrix norm, selecting the $k$ largest singular values leads to an optimal approximation of $\mathbf{W}_l$ under rank constraints (Mirsky, 1960). It has been previously argued, however, that this approach does not yield satisfactory results in deep learning as is does not take into account the underlying (training) data and downstream task (Hsu et al., 2022). Different approaches have been presented to address this problem (Hsu et al., 2022; Yuan et al., 2024; Wang et al., 2024; Chen et al., 2021).

### 2.1.3 $\ell_1$-Regularization

Consider a generic loss functional $\mathcal{L}(\mathbf{X}; \boldsymbol{\theta})$ evaluated on some model that is specified by a parameter vector $\boldsymbol{\theta}$ and data $\mathbf{X}$. During model training, sparsity in $\boldsymbol{\theta}$ is typically encouraged by solving the penalized optimization problem

$$\min_{\boldsymbol{\theta}} \mathcal{L}(\mathbf{X}; \boldsymbol{\theta}) + \lambda \left\| \boldsymbol{\theta} \right\|_1, \quad \lambda > 0, \tag{5}$$

which is known as $\ell_1$-*regularization*, or *least absolute shrinkage and selection operator (LASSO)* in case of (generalized) linear models (Tibshirani, 1996; Hastie et al., 2015). Feature selection plays a key role in this optimization problem. Indeed, increasing $\lambda$ in (5) leads to sparser solutions $\hat{\boldsymbol{\theta}}(\lambda)$. This effectively gives rise to a feature importance ranking for the solution of (5) through the so-called regularization path (Tibshirani, 1996; Efron et al., 2004; Mairal & Yu, 2012) — an idea that inspired the approach of `ACIP`. Here, 'feature importance' is understood as the relative contribution to the (task- and data-specific) loss $\mathcal{L}(\mathbf{X}; \boldsymbol{\theta})$, i.e., a feature is considered more important if its removal causes a larger increase in the loss. In particular, different choices of $\mathcal{L}$ may lead to different solution paths and feature rankings. Furthermore, previous results indicate that optimization paths of iterative schemes can be related to regularization paths (Suggala et al., 2018). Intuitively, this suggests that general optimization paths of iterative schemes for $\ell_1$-objectives reveal information about feature importance in the context of sparse models as well.

In `ACIP`, we will successively increase $\lambda$ to introduce a higher degree of model compression in terms of parameter sparsity in a controlled manner (see also Remark 2.1 below).

## 2.2 Any Compression via Iterative Pruning (`ACIP`)

Algorithmically, `ACIP` consists of the following three key steps, which are detailed in the sections below. For a schematic visualization of `ACIP`, we refer to Figure 2.

**Step 1. (Model Reparametrization)** Apply SVD to the weights of all (dense) linear layers according to (6). Introduce low-rank adapters $\mathbf{\Delta}$ and singular value masks as tunable parameters.

**Step 2. (Scoring via Iterative Pruning)** Choose a surrogate loss $\mathcal{L}$ and a calibration data set $\mathbf{X}$. Perform iterative pruning of singular value masks $\mathbf{p}$ and simultaneous tuning of low-rank adapters $\mathbf{\Delta}$ by applying $\ell_1$-regularized gradient-based optimization as in (8). Obtain a global parameter score map $\boldsymbol{\rho}$ by using Algorithm 1.

**Step 3. (Any Compression)** Choose *any* desired compression rate. Use the score map $\boldsymbol{\rho}$ and low-rank adapters $\mathbf{\Delta}$ to materialize the compressed model in real-time.

### 2.2.1 Step 1. Model Reparametrization

We start by reparametrizing all linear layers of a network[2] using SVD as described in (3) and assign

$$\mathbf{W}_l \leftarrow \mathbf{U}_l \mathbf{M}_l \mathbf{\Sigma}_l \mathbf{V}_l^\top + \mathbf{\Delta}_l, \tag{6}$$

where $l$ denotes the layer index, $\mathbf{M}_l$ is a diagonal matrix with binary entries $\mathbf{m}_l^{(i)} \in \{0, 1\}$ masking the singular values $\mathbf{s}_l^{(i)}$ in $\mathbf{\Sigma}_l$, and $\mathbf{\Delta}_l$ is a low-rank adapter (LoRA) (Hu et al., 2022). In all subsequent steps of `ACIP`, we freeze $\mathbf{\Sigma}_l$, $\mathbf{U}_l$, and $\mathbf{V}_l$.

We find that adding a low-rank adapter helps to compensate for potential errors that are introduced by pruning in Step 2 (Section 2.2.2). We initialize $\mathbf{M}_l$ as the identity matrix and $\mathbf{\Delta}_l$ as zero weights. In this way, the reparametrized model remains identical to the original model up to numerical precision.

We assign the binary masks $\mathbf{m}_l^{(i)}$ such that $\tilde{\mathbf{s}}_l^{(i)} = \mathbf{m}_l^{(i)} \cdot \mathbf{s}_l^{(i)}$ represents the pruned or retained singular values, respectively. Thus, $\mathbf{m}_l^{(i)}$ decouples the magnitude of a singular value and the pruning decisions based on its importance.

The above parametrization leads to a parameter-efficient compression scheme. Indeed, given an $m \times n$ matrix $\mathbf{W}$, the number of non-zero singular values is bounded by $r = \min(m, n)$, which means the number of tunable mask parameters scales linearly in the feature dimensions.

**Mask parametrization.** We parametrize the binary masks through a thresholding operation of the form

$$\mathbf{m}_l^{(i)} = \begin{cases} 0, & \text{for } \mathbf{p}_l^{(i)} \leq 0 \\ 1, & \text{for } \mathbf{p}_l^{(i)} > 0 \end{cases}, \tag{7}$$

where $\mathbf{p}_l^{(i)}$ are scalar learnable parameters. As this operation is not differentiable, we use the straight-through estimator for backpropagation (Bengio et al., 2013; Yin et al., 2018).

### 2.2.2 Step 2. Scoring via Iterative Pruning

We now aim to build a global score map over all singular values in the reparametrized layers, which guide model compression subsequently in Step 3 (Section 2.2.3). Leveraging the sparsity-inducing property of $\ell_1$-regularization (see Section 2.1.3), we progressively shrink the mask parameters $\mathbf{p}_l^{(i)}$ to zero and derive a score map based on the pruning order. The two key algorithmic components of this "iterative scoring" strategy are presented next.

**Iterative pruning.** The optimization problem solved by `ACIP` takes the form

$$\min_{\mathbf{p}, \mathbf{\Delta}} \mathcal{L}(\mathbf{X}; \boldsymbol{\theta}, \mathbf{p}, \mathbf{\Delta}) + \lambda \|\mathbf{p}\|_1, \tag{8}$$

where $\mathcal{L}$ denotes a suitable calibration loss for the model, $\mathbf{p} = \{\mathbf{p}_0, \ldots, \mathbf{p}_L\}$ is the set of all mask parameters, $\mathbf{\Delta} = \{\mathbf{\Delta}_0, \ldots, \mathbf{\Delta}_L\}$ is the set of all low-rank adapters, and $\boldsymbol{\theta}$ is the set of all remaining model parameters that

---

[2]Following common practice, we ignore the embedding layer and classification head in (decoder-only) transformers.

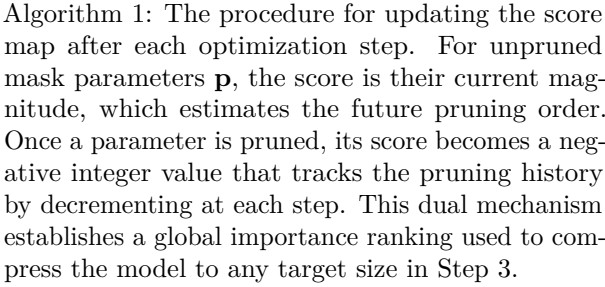

```
1  # params: parameters p, vectorized;
2  # scores: pruning scores, vectorized;
3  #         initialized to NaN
4
5  def update_scores(scores, params):
6    # previously pruned parameters
7    score_mask = scores <= 0.
8    scores[not score_mask] = params[not
         score_mask]
9    scores[score_mask] -= 1.
10   return scores
```

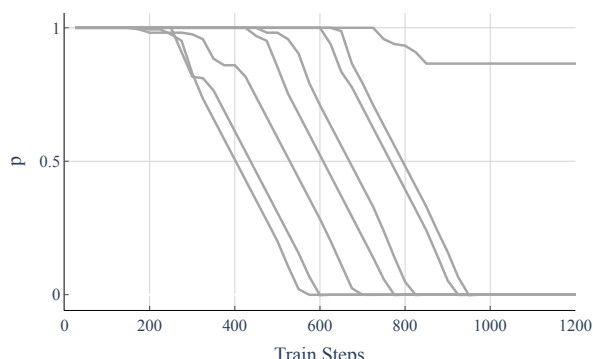

Figure 3: Progressive shrinkage of exemplary mask parameters $\mathbf{p}$ in `Attn-V` layer $l = 30$ of **LLaMA-7B** based on (8). Each plotted line corresponds to the evolution of a parameter value over training time. The starting points of shrinkage are predictive of the pruning order, a typical phenomenon in $\ell_1$-regularization. In `ACIP`, this pruning order determines the score of associated singular values $\mathbf{s}$ (cf. Algorithm 1).

Algorithm 1: The procedure for updating the score map after each optimization step. For unpruned mask parameters $\mathbf{p}$, the score is their current magnitude, which estimates the future pruning order. Once a parameter is pruned, its score becomes a negative integer value that tracks the pruning history by decrementing at each step. This dual mechanism establishes a global importance ranking used to compress the model to any target size in Step 3.

are frozen during optimization. We perform gradient-based optimization until a preset maximum compression ratio $r_{\text{stop}}$ is reached (see Appendix D.4 for further discussion). Optionally, we perform post-tuning for a fixed number of steps by freezing the masks $\mathbf{p}$ and continuing the optimization of the low-rank adapters $\boldsymbol{\Delta}$.

*Remark* 2.1 (Scaling of $\lambda$). If $\lambda$ is chosen too small, the maximum compression ratio $r_{\text{stop}}$ might never be reached. If $\lambda$ is too large, training might become unstable and the score map ambiguous. Therefore, we use a simple linear scheduler that scales $\lambda$ by a fixed factor $>1$ every $j$ optimization steps, so that pruning becomes increasingly aggressive over time.

*Remark* 2.2 (Role of the Calibration Loss). As pointed out in Section 2.1.3, a feature importance ranking as produced by optimizing (8) depends on the specific loss $\mathcal{L}(\mathbf{X}; \boldsymbol{\theta}, \mathbf{p}, \boldsymbol{\Delta})$. Given a model to be compressed, this leaves two important design decisions to the user of `ACIP`, namely the choice of loss function $\mathcal{L}$ and calibration data. In the case of LLMs, it is natural to use the same loss function as in pre-training, i.e., the negative log-likelihood loss for next-token prediction. The calibration data should be representative of the task for which the compressed model(s) should perform well. This choice is therefore very use-case dependent and could range from very specific fine-tuning tasks to general-purpose prediction. In this work, we focus on the latter.

**From iterative pruning to score map.** The optimization process of (8) is used to construct our score map. Based on the discussion of Section 2.1.3, we hypothesize that there is a close relationship between the order in which the parameters $\mathbf{p}_l^{(i)}$ vanish and their importance for the model — the least important parameters are pruned first and so on. When conducting our experiments, we observed a shrinkage behavior that supports this hypothesis; see Figure 3 for a specific example.

Algorithm 1 describes how the score map is updated after each optimization step to represent feature importances. In plain words, the score map is built based on the pruning order. A negative number in the map indicates how many steps ago a parameter was pruned. For all parameters that have not been pruned, the score is set to the value of the corresponding parameter. We refer to Appendix D.9 for visual examples of `ACIP` score maps.

The approach of Algorithm 1 ensures that (i) the score map stores the pruning history, and (ii) it estimates future pruning based on the parameter magnitudes. Note that absolute values of the score are irrelevant for parameter ranking.

### 2.2.3 Step 3. Any Compression

From Step 2, we only retain the score map $\boldsymbol{\rho}$ and the low-rank adapters $\boldsymbol{\Delta}$. In particular, the pruned masks $\mathbf{m}_l^{(i)}$ are discarded, as they are irrelevant for compression at this stage (cf. Figure 2). As motivated above, the score map allows us to globally rank all singular values based on their score. This leads to a fully independent compression stage where we can flexibly create a model of any reduced size: we prune as many singular values $\mathbf{s}_l^{(i)}$ (and the corresponding singular vectors) according to their scores $\boldsymbol{\rho}_l^{(i)}$ so that a given compression rate $r$ is achieved. Note that there is a monotonic but non-linear relationship between the total number of pruned singular values $k$ and compression rate $r$ (see Section 2.1.2). Given a target rate $r$, we find $k$ via a binary search (in almost real-time). As this compression procedure operates directly on the reparametrized model from Step 1, it is reversible and therefore indeed allows for Any Compression; for a slightly refined version, see Appendix D.3.

Finally, to materialize a model at a fixed target rate, all pruned singular values and vectors are discarded, so that the initial SVD-reparametrization turns into an actual low-rank factorization. For layers with determined rank $k \geq \frac{mn}{m+n}$, i.e., where a factorization would not save any parameters, we avoid an inefficient storage usage by simply recovering the (dense) weight matrix from its SVD components.

*Remark* 2.3 (Merging Low-Rank Adapters). Above, we have treated the truncated weight matrices $\tilde{\mathbf{W}}_l$ and the corresponding low-rank adapters $\boldsymbol{\Delta}_l$ as distinct components. It is possible to merge the two decompositions into a single one to improve model throughput. To this end, let $\boldsymbol{\Delta}_l = \mathbf{A}_l \mathbf{B}_l^\top$ be the factorization of the low-rank adapter. Then, we can express the reparametrization at layer $l$ in (6) by matrix concatenations of the form

$$\mathbf{W}_l = \begin{bmatrix} \mathbf{U}_l \boldsymbol{\Sigma}_l, \mathbf{A}_l \end{bmatrix} \begin{bmatrix} \mathbf{V}_l^\top \\ \mathbf{B}_l^\top \end{bmatrix}. \tag{9}$$

After specifying a compression rate, the corresponding weight matrix can be expressed as

$$\tilde{\mathbf{W}}_l = \underbrace{\begin{bmatrix} \tilde{\mathbf{U}}_l \tilde{\boldsymbol{\Sigma}}_l, \mathbf{A}_l \end{bmatrix}}_{=:\mathbf{M}_2} \underbrace{\begin{bmatrix} \tilde{\mathbf{V}}_l^\top \\ \mathbf{B}_l^\top \end{bmatrix}}_{=:\mathbf{M}_1}, \tag{10}$$

where $\tilde{\mathbf{U}}_l, \tilde{\boldsymbol{\Sigma}}_l, \tilde{\mathbf{V}}_l^\top$ are the truncated singular components, see (4). Note that $\tilde{\mathbf{W}}_l$ is never explicitly materialized as a dense matrix in memory (to ensure actual memory savings) and its forward pass is implemented as two consecutive vector-matrix multiplications with $\mathbf{M}_1$ and $\mathbf{M}_2$.

### 2.3 Computational Considerations

In this section, we discuss the computational costs and memory overhead of `ACIP` and compare them to other common compression paradigms. As detailed in Section 2.2, the overall process of `ACIP` can be divided into three distinct stages, each with different computational characteristics. A summary of the empirical runtime and memory costs for a LLaMA-7B base model is provided in Table A5.

The most resource-intensive stage of `ACIP` is scoring via iterative pruning (Step 2). It involves backpropagation through the entire model to update the singular value masks and the low-rank adapters. As such, these updates are parameter-efficient, but the overall memory consumption is still proportional to the full model size, which can be demanding. In contrast, backpropagation-free, layer-wise methods such as ASVD (Yuan et al., 2024) and SVD-LLM (Wang et al., 2024) are notably more memory-efficient, as they only need to process and store the data for a single layer at any given time. However, it is crucial to emphasize that the iterative pruning stage is compatible with Any Compression and represents a one-time, upfront computational investment. This calibration can be performed once by a model provider, amortizing the cost across all subsequent uses. Another noteworthy aspect of Step 2 is that the SVD-reparametrization initially leads to an increased model size compared to the original model (cf. Table A5), since the singular vector matrices $\mathbf{U}$ and $\mathbf{V}$ both need to be stored in memory.

Once the score map is generated, the compression stage (Step 3) is exceptionally efficient. As shown in Table A5, compressing the model to any target size by discarding singular vectors based on their global scores

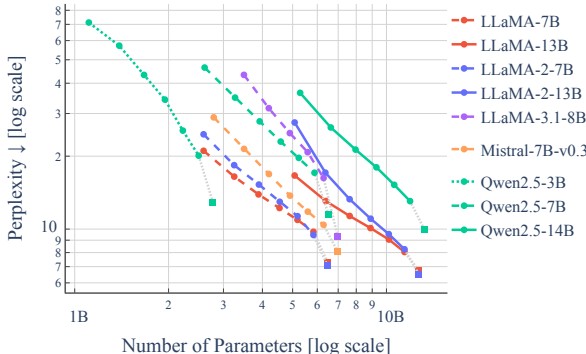
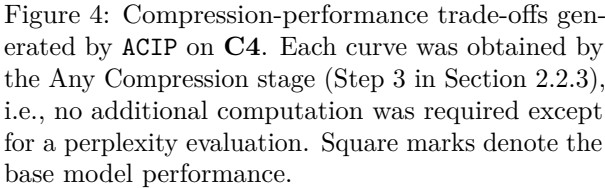
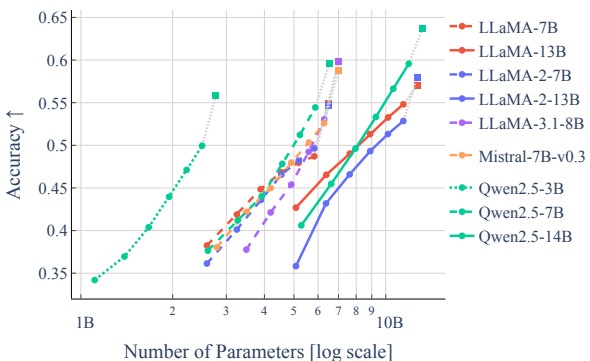

Figure 4: Compression-performance trade-offs generated by `ACIP` on **C4**. Each curve was obtained by the Any Compression stage (Step 3 in Section 2.2.3), i.e., no additional computation was required except for a perplexity evaluation. Square marks denote the base model performance.

Figure 5: Compression-performance trade-off curves generated by `ACIP`, using average accuracy on all **LM-Eval** tasks as metric.

occurs in near real-time. This allows practitioners to dynamically select the optimal compression-performance trade-off for their specific application without incurring any significant computational delay.

## 3    Experiments

**Experimental setup.**    To demonstrate effectiveness across architectural differences in LLMs, we evaluate `ACIP` on a selection of popular open-weight models: LLaMA-7B/13B (Touvron et al., 2023a), LLaMA-2-7B/13B (Touvron et al., 2023b), LLaMA-3.1-8B (Grattafiori et al., 2024), Qwen2.5-7B/14B (Qwen et al., 2024), and Mistral-7B-v0.3 (Jiang et al., 2023). We use a subset of C4 training data (Raffel et al., 2019) for the pruning stage. Regarding evaluation tasks, we follow Wang et al. (2024) and report perplexity on validation held-outs of C4 (Raffel et al., 2019) and WikiText-2 (Merity et al., 2017), and we consider seven zero-shot tasks from EleutherAI LM Evaluation Harness (LM-Eval) (Gao et al., 2023). More implementation details about `ACIP` and choices of hyperparameters can be found in Appendix B.

### 3.1    Analyzing Compression-Performance Trade-Offs

We first study compression-performance trade-offs powered by `ACIP`. Figures 4 and 5 demonstrate smooth and consistent curve shapes for all considered models; analogous results for WikiText-2 and individual zero-shot LM-Eval tasks can be found in Figure A8. We note that a monotonic relationship between size and performance is not self-evident, e.g., see Figure A13 in Appendix D.6 for a trivial approach that uses magnitude-based pruning. More insights on the runtime and memory consumption of `ACIP` as well as inference speed of compressed models are reported in Table A5 in Appendix D.1.

A remarkable observation is that the oldest models, LLaMA-7B/13B, perform best perplexity-wise, while newer, more capable models like Qwen2.5-7B/14B dominate on LM-Eval as expected, especially on the lower compression levels. This apparent contradiction is likely caused by a deviation of the pre-training data distributions from C4 in the case of more recent models.

A second noteworthy outcome of Figures 4 and 5 are the gaps between LLMs of different base model sizes in the same family. Indeed, `ACIP` cannot match the performance of base models of smaller size, e.g., compare the compressed Qwen2.5-7B with the original Qwen2.5-3B. This is not surprising because the corresponding smaller-size base models were obtained by pre-training or knowledge distillation (Hinton et al., 2015; Busbridge et al., 2025), which are orders of magnitudes more expensive than `ACIP`.

Table 1: *Any Compression under SVD reparameterization.* Zero-shot evaluation of **LLaMA-7B**. Comparison with baselines SVD magnitude pruning (SVD-Magn.), ASVD (Yuan et al., 2024), SVD-LLM (Wang et al., 2024), and Dobi-SVD (without remapping) (Qinsi et al., 2025). ↑: larger is better; ↓: smaller is better; best results for each task and size ratio are marked in **bold**. The scores for ASVD and SVD-LLM are taken from Wang et al. (2024) and the scores for Dobi-SVD from (Qinsi et al., 2025).

| Size | Method | C4 ↓ | WikiText-2 ↓ | Openb. ↑ | ARC_e ↑ | WinoG. ↑ | HellaS. ↑ | ARC_c ↑ | PIQA ↑ | MathQA ↑ | LM Eval Avg. ↑ |
|---|---|---|---|---|---|---|---|---|---|---|---|
| 100% | Original | **7.34** | **5.68** | **0.28** | **0.67** | **0.67** | **0.56** | **0.38** | **0.78** | **0.27** | **0.52** |
| 80% | SVD-Magn. | 9819.35 | 11312.47 | 0.15 | 0.27 | 0.49 | 0.26 | 0.20 | 0.53 | 0.20 | 0.30 |
| | ASVD | 15.93 | 11.14 | 0.25 | 0.53 | 0.64 | 0.41 | 0.27 | 0.68 | **0.24** | 0.43 |
| | SVD-LLM | 15.84 | **7.94** | 0.22 | 0.58 | 0.63 | 0.43 | 0.29 | 0.69 | **0.24** | 0.44 |
| | Dobi-SVD | **10.01** | 8.54 | 0.26 | 0.59 | **0.66** | 0.44 | 0.31 | 0.70 | 0.23 | 0.46 |
| | ACIP (ours) | 10.92 | 8.83 | **0.28** | **0.66** | 0.63 | **0.49** | **0.32** | **0.74** | 0.23 | **0.48** |
| 70% | SVD-Magn. | 15186.30 | 12575.68 | 0.15 | 0.26 | 0.51 | 0.26 | 0.22 | 0.52 | 0.20 | 0.30 |
| | ASVD | 41.00 | 51.00 | 0.18 | 0.43 | 0.53 | 0.37 | 0.25 | 0.65 | 0.21 | 0.38 |
| | SVD-LLM | 25.11 | **9.56** | 0.20 | 0.48 | 0.59 | 0.37 | 0.26 | 0.65 | 0.22 | 0.40 |
| | ACIP (ours) | **12.22** | 10.35 | **0.28** | **0.64** | **0.62** | **0.47** | **0.31** | **0.73** | **0.23** | **0.47** |
| 60% | SVD-Magn. | 14414.87 | 20987.82 | 0.15 | 0.25 | 0.50 | 0.25 | 0.22 | 0.52 | 0.19 | 0.30 |
| | ASVD | 1109.00 | 1407.00 | 0.13 | 0.28 | 0.48 | 0.26 | 0.22 | 0.55 | 0.19 | 0.30 |
| | SVD-LLM | 49.83 | 13.11 | 0.19 | 0.42 | 0.58 | 0.33 | 0.25 | 0.60 | 0.21 | 0.37 |
| | Dobi-SVD | 23.54 | 13.54 | 0.22 | 0.41 | 0.58 | 0.34 | 0.27 | 0.61 | 0.23 | 0.38 |
| | ACIP (ours) | **13.91** | **12.46** | **0.25** | **0.61** | **0.59** | **0.44** | **0.30** | **0.71** | **0.24** | **0.45** |
| 50% | SVD-Magn. | 62899.32 | 109019.52 | 0.15 | 0.26 | 0.50 | 0.26 | 0.23 | 0.54 | 0.18 | 0.30 |
| | ASVD | 27925.00 | 15358.00 | 0.12 | 0.26 | 0.51 | 0.26 | 0.22 | 0.52 | 0.19 | 0.30 |
| | SVD-LLM | 118.57 | 23.97 | 0.16 | 0.33 | 0.54 | 0.29 | 0.23 | 0.56 | 0.21 | 0.33 |
| | ACIP (ours) | **16.47** | **16.16** | **0.21** | **0.57** | **0.57** | **0.40** | **0.27** | **0.68** | **0.22** | **0.42** |
| 40% | SVD-Magn. | 29804.73 | 29364.55 | 0.14 | 0.27 | 0.51 | 0.26 | 0.22 | 0.53 | 0.20 | 0.30 |
| | ASVD | 43036.00 | 57057.00 | 0.12 | 0.26 | 0.49 | 0.26 | 0.21 | 0.51 | 0.18 | 0.29 |
| | SVD-LLM | 246.89 | 42.30 | 0.14 | 0.28 | 0.50 | 0.27 | 0.22 | 0.55 | 0.21 | 0.31 |
| | Dobi-SVD | 190.62 | 46.18 | 0.15 | 0.31 | 0.52 | 0.28 | 0.20 | 0.54 | **0.22** | 0.32 |
| | ACIP (ours) | **21.05** | **23.99** | **0.19** | **0.49** | **0.55** | **0.35** | **0.24** | **0.64** | 0.21 | **0.38** |

## 3.2 Comparison to Existing Works

We now compare `ACIP` to recent works focusing on SVD-based structured pruning, namely ASVD (Yuan et al., 2024), SVD-LLM (Wang et al., 2024), and Dobi-SVD (without remapping) (Qinsi et al., 2025). The former two approaches are backpropagation-free and perform (activation-aware) layer-wise updates instead, while Dobi-SVD proposes a differentiable truncation mechnism for singular values. Moreover, we evaluated a simple SVD magnitude pruning approach (SVD-Magn.), where we set the score map equal to the singular values of the weight matrices. This technique allows for Any Compression analogously to `ACIP` and therefore serves as another natural baseline; see Appendix D.6 for further study.

Table 1 shows that `ACIP` consistently outperforms all baseline methods with a growing gap for higher compression levels. Note that SVD-LLM and Dobi-SVD were calibrated on WikiText-2 instead of C4, which might explain slightly better results on the former dataset for 70% and 80% size. We think that these results underpin the benefits of an end-to-end scheme: (i) a simultaneous correction, e.g., by LoRA, can drastically improve performance, and (ii) robust pruning patterns can be found without leveraging any specific features of the SVD factorization. Moreover, we note that re-computations are required to generate each row of Table 1 for ASVD, SVD-LLM, and Dobi-SVD, whereas `ACIP` only needs a single run. Analogous results for `ACIP` applied to all other models can be found in Table A2.

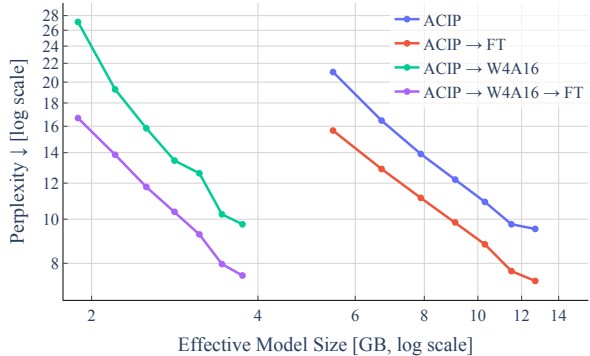
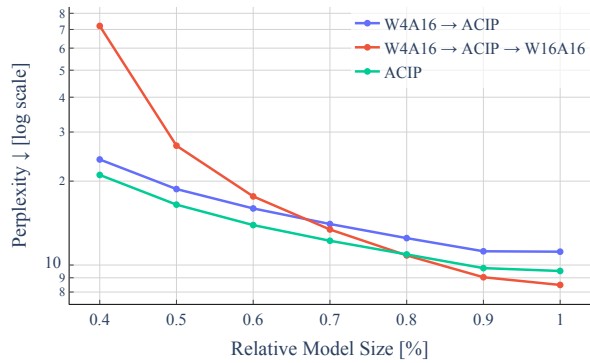

Figure 6: Compression-performance trade-off curves for **LLaMA-7B** on **C4** showing the impact of **fine-tuning** and **quantization** *after* compression with `ACIP`. The horizontal axis measures size in terms of required (weight) memory to visualize the gains of quantization more clearly.

Figure 7: Compression-performance trade-off curves for **LLaMA-7B** on **C4**, showing that **quantization** *before* `ACIP` leads to similar results as without.

### 3.3 Improving Performance Through Fine-Tuning

While the main goal of this work is to produce a full family of accurate, compressed models from a few optimization steps, their performance can be certainly improved through continued fine-tuning. Figure 6 highlights the gains of fine-tuning LLaMA-7B; see Table A2 for more detailed numerical results on all other models. We observe that fine-tuning leads to a performance offset that is almost constant across all compression levels, which underlines the predictive capacity of `ACIP`. Note that we even observe a jump at zero compression because inserting the low-rank adapters learned by `ACIP` leads to a slight initial performance drop (see Appendix D.3 for a potential improvement).

An optional fine-tuning step is not exclusive to `ACIP` but can be applied to many other compression approaches as well. Table A3 provides a comparison with ASVD (Yuan et al., 2024) and SVD-LLM (Wang et al., 2024) (cf. Section 3.2) when fine-tuned with LoRA. While `ACIP` still performs best in this respect, we argue that post-compression fine-tuning should be still seen as an independent (and much more costly) algorithmic step for two reasons. (i) Its outcome strongly depends on the specific training protocol and data, making a fair and direct comparison challenging; (ii) it requires us to fix a compression level, which breaks the crucial Any Compression feature of `ACIP`. Therefore, promoting a costly fine-tuning step after compression is not the primary concern of our work.

### 3.4 Combining `ACIP` with Quantization

In the field of low-cost compression for LLMs, quantization is still considered as the gold standard (Hohman et al., 2024; Zhu et al., 2024), so that a practitioner might not be willing to exchange its gains for the benefits of `ACIP`. Fortunately, `ACIP` only tunes a tiny fraction of weights with high precision, so that all remaining modules are suitable for quantization. In our experiments, we quantize all parameterized and unparametrized linear layers to 4-bit in `fp4`-format (Dettmers et al., 2023) using the `bitsandbytes`-Package (W4A16), except for the embedding layer and final classification head. We study the gains of quantization for `ACIP` in the following two ways.

**Compress first, then quantize.** We first apply `ACIP` as usual, compress the model to a given target size, and then quantize all linear layers. Figure 6 confirms that this approach works fairly well, only producing a slight performance drop compared to non-quantized versions; see Table A4 for a full evaluation on all other metrics. We also observe that an optional fine-tuning step as in Section 3.3 can almost fully compensate for the errors introduced by quantization after compression. This finding is well in line with the effectiveness of

the popular QLoRA approach (Dettmers et al., 2023). Moreover, Figure 6 reveals a drastic improvement through quantization in terms of required memory. Here, the `ACIP`-trade-off allows practitioners to study and apply a more fine-grained compression on top of quantization.

**Quantize first, then compress and transfer.** Compared to layer-wise methods like ASVD and SVD-LLM, `ACIP` has a higher demand in GPU memory due to backpropagation. A quantization of all frozen weight matrices can be an effective remedy in this respect. For the experiment shown in Figure 7, we have applied quantization before `ACIP`, which leads to very similar compression-performance trade-offs as in the non-quantized case. Going one step further, we transfer the score maps and low-rank adapters from this quantized version of `ACIP` back to full precision: We load the base model in `bf16`, apply layer-wise SVD-parametrization, insert the low-rank adapters learned by quantized `ACIP`, and use the corresponding score map to obtain a compressed model (W16A16). The resulting trade-off curve in Figure 7 confirms that this simple strategy works fairly well, especially for lower compression levels.

### 3.5 Further Experiments and Ablations

Several additional experiments are presented in Appendix D, analyzing the impact of several key components and design aspects of `ACIP`. Starting with an analysis of algorithmic efficiency and latency (Appendix D.1), we study the impact of the low-rank adapters (Appendix D.2), compression rule (Appendix D.3), stopping criterion (Appendices D.4 and D.5), score map design (Appendix D.6), post-tuning (Appendix D.7), and specific types of linear layers (Appendix D.8). Finally, we show examples of score maps (Appendix D.9) and prompt completions by compressed models (Appendix D.10).

## 4 Related Work

In this section, we pick up on our broader discussion on the field of model compression from Section 1 and put our work in context with several directly related branches of research.

**Structured pruning & low-rank factorization.** Conceptually, `ACIP` falls under the umbrella of *structured* parameter pruning, specifically, low-rank matrix decomposition. The rationale behind this compression approach is to approximate large weight matrices by products of low-rank factors to reduce the total parameter count, and at the same time, to preserve critical information. After initial efforts into this direction for smaller language models (Edalati et al., 2022; Tahaei et al., 2022), techniques for LLMs primarily built on (weighted) SVD of linear layers (Ben Noach & Goldberg, 2020; Hsu et al., 2022).

However, a key challenge of SVD-based pruning is that simply truncating singular values based on magnitude alone is insufficient and makes additional fine-tuning on downstream tasks necessary. Follow-up work recognized that the poor approximations are caused by LLM weights being high-rank and instead turned to decomposing network features which are sparse (Kaushal et al., 2023; Yu & Wu, 2023). Similarly, recent studies (Sharma et al., 2023; Yuan et al., 2024; Jaiswal et al., 2024) have shown rank reduction to differently affect layers in a network and proposed heuristics for non-uniform pruning. Going even further, ASVD (Yuan et al., 2024) pointed out the importance of activation-aware approximations, proposing a training-free compression method that takes the (calibration) data distribution into account. Building on this, SVD-LLM (Wang et al., 2024) recently derived an analytical layer-wise correction leading to superior compression results. While `ACIP` promotes an activation-aware solution as well, it relies on gradient-based optimization, which avoids any SVD-specific feature engineering and allows for simultaneous errors corrections.

Another relevant line of work on structured pruning aims to jointly remove groups of parameters or entire network components, e.g., weight matrix columns/rows, network layers, or attention heads (Frantar & Alistarh, 2023; Ma et al., 2023; Xia et al., 2024; Ashkboos et al., 2024; Kim et al., 2024). At high compression rates, however, such "coarse" approaches often remove critical substructures, causing a significant performance drop that is only recoverable through additional fine-tuning.

**Score maps & Any Compression.** A common feature of the aforementioned structured pruning approaches is that they first truncate parameters to a preset target size and then compute an error correction.

This design choice means that exploring the full compression-performance trade-off requires repeated, costly computations for each desired compression ratio (cf. Figure 1(a)). In contrast, `ACIP` overcomes this limitation by using score maps to determine global parameter importance. As such, score maps have been used as a tool for model compression since the 1980s (LeCun et al., 1989; Hassibi et al., 1993). However, in the era of LLMs, deriving a score for each parameter poses significant challenges in terms of scalability. Addressing this concern, `ACIP` enables scalable Any Compression by (i) using weight factorization to significantly reduce the score map size (e.g., to ∼900k parameters for a 7B model), and (ii) decoupling the scoring and compression stages (Step 2 and 3 in Section 2.2, respectively).

**Any-size pre-training.** Beyond post-training compression, an alternative paradigm for obtaining models of varying sizes is to incorporate this flexibility into the (pre-)training process itself. For example, Cai et al. (2020) train a single, large "Once-for-all" network from which specialized sub-networks of different sizes can be extracted without retraining. More recently, the MatFormer (Devvrit et al., 2024) has demonstrated how to build a family of "any-size" models directly through a nested pre-training methodology, based on Matryoshka Representation Learning (Kusupati et al., 2022). The recently published Gemma-3n model uses this approach in production to make LLMs ready for mobile and edge devices (Gonzalez & Shivanna, 2025). While these methods also produce a trade-off between model size and performance, they require a significantly higher upfront computational budget associated with complex pre-training from scratch. `ACIP` provides a lightweight, post-training alternative that offers similar flexibility for any existing pre-trained model.

**Rate-distortion theory.** Finally, our work can be viewed through the lens of rate-distortion theory, which investigates the analytical trade-off between achievable data compression rates and the error (distortion) introduced by lossy compression (Cover & Thomas, 2006). While some recent work (Gao et al., 2019; Isik et al., 2022) investigates rate-distortion theory of machine learning models for simple architectures under rather specific assumptions, the information-theoretic limits of neural network compression are generally unknown in practically relevant settings. In this context, the family of compressed models generated by `ACIP` conveniently provides an empirical (upper) bound on the distortion-rate function of a large-scale model from a single optimization run.

## 5 Conclusion

In this work, we have introduced *Any Compression via Iterative Pruning* (`ACIP`), a simple end-to-end algorithm to determine the compression-performance trade-off of pre-trained models. The underlying score map ranking allows us to materialize models of any compression rate in real-time. We have demonstrated empirically that the downstream performance of the resulting models is superior to existing, layer-wise factorization approaches. The flexibility and efficiency of `ACIP` make it a practical tool for deploying large-scale models in resource-constrained settings, especially in combination with other compression techniques such as quantization.

**Discussion.** Our main results in Figures 4 and 5 resemble the well-known phenomenon of scaling laws (Kaplan et al., 2020; Hoffmann et al., 2022). Recently, it has been shown that any-size models can be achieved through pre-training (Devvrit et al., 2024; Gonzalez & Shivanna, 2025) (see also Section 4), exhibiting similar trade-offs as `ACIP`. Establishing a rigorous connection between these two fields of research could be a fruitful avenue of future work.

In a similar vein, we observe that more recent models tend to be less compressible (e.g., compare the slopes of LLaMA-13B and Qwen2.5-14B in Figure 5). We hypothesize that this relates to newer models carrying denser information per weight, since they were trained on much larger datasets (Allen-Zhu & Li, 2024). Also, the distribution of the calibration dataset (C4 in our case) might play an important role in this context.

A notable technical limitation of our work is that we have only focused on models that are tunable on a single (NVIDIA H100) GPU in `bf16`-precision. Hence, the scaling behavior of `ACIP` for larger LLMs (30B+) remains to be explored. We also emphasize that `ACIP` could be transferred to other modalities, architectures, and tasks without any notable modifications. Finally, a more detailed study of inference speed (beyond the results of Appendix D.1) could provide useful insights into the interplay of low-rank models and their efficiency.

**Broader Impact Statement**

The primary broader impact of our work lies in increasing the accessibility and practicality of large Foundation Models. By empowering practitioners to effortlessly navigate the compression-performance trade-off without costly recomputation, `ACIP` helps to democratize the deployment of advanced AI. This could unlock novel applications in resource-scarce domains — from on-device mobile assistants to intelligent systems in manufacturing and automotive sectors — and support researchers with limited computational budgets. Ultimately, our approach contributes to a more sustainable and inclusive AI ecosystem by enabling the efficient use of pre-trained models, reducing both computational and environmental overhead.

In parallel with these benefits, it is crucial to acknowledge the ethical implications and responsibilities that come with democratizing powerful technology. The same accessibility that fosters innovation could also lower the barrier for malicious applications, such as the efficient generation of spam or real-time misinformation on edge devices. This underscores the critical need for the AI/ML community to proactively develop robust safety mechanisms and ethical guidelines that are effective for models of all sizes. By doing so, we can ensure the positive impacts of accessibility are not undermined by potential misuse.

**Software and Data**

Code is available under `https://github.com/merantix-momentum/acip`. ACIP-Models are available under `https://huggingface.co/collections/MerantixMomentum/acip`.

**Acknowledgments**

The authors thank Brennan Wilkerson and Ziyad Sheebaelhamd for helpful discussions. We thank John Arnold and Jannis Klinkenberg for their technical cluster support.

We kindly acknowledge funding by the German Federal Ministry of Education and Research (BMBF) within the project "More-with-Less: Effiziente Sprachmodelle für KMUs" (grant no. 01IS23013A). Computational resources were provided by the German AI Service Center WestAI and used to conduct the numerical experiments of this work.

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

## Contents of Appendix

# A    Additional Remarks

In this section, we discuss a few additional aspects (in Q&A format) about our method and experimental design that were not (fully) addressed in the main part for the sake of brevity.

Q1. *Why did you not directly compare your results to quantization and full-weight (unstructured) pruning?*

A1. We argue that these are fundamentally different compression approaches. Full weight manipulations, in principle, have the potential to lead to more powerful compressions because they have more degrees of freedom (analogously to full-weight fine-tuning vs. PEFT). Therefore, they should not be seen as competing methods but complementary ones. We admit that practitioners probably would not favor `ACIP` over well-established and widely supported quantization techniques. However, the adapter-style nature of `ACIP` makes it suitable for a combination. This can, for example, allow `ACIP` to be further improved through a quantization of the singular vector matrices as demonstrated in Section 3.4.

Q2. *Why did you not compare with model distillation or combine `ACIP` with it?*

A2. While model distillation can lead to outstanding compression results, e.g., see (Busbridge et al., 2025; Raschka, 2024), this approach requires significantly more resources than `ACIP`, typically orders of magnitudes more. A direct comparison is therefore not meaningful from our point of view, as it should at least be based on approximately the same computational budget.

Q3. *Why do you propose a backpropagation-based algorithm instead of layer-wise weight updates?*

A3. Let us first summarize several benefits of our end-to-end optimization approach from the main paper: (i) it is conceptually simple and requires no feature engineering, (ii) an error correction can be injected with almost no extra costs, (iii) it allows us to perform efficient and accurate Any Compression.

Apart from that, and to the best of our knowledge, existing compression algorithms that use layer-wise updates like ASVD (Yuan et al., 2024), SVD-LLM (Wang et al., 2024), or WeLore (Jaiswal et al., 2024) require a separate fine-tuning step to achieve competitive downstream performance at stronger compression ratios. Therefore, the lower costs of layer-wise compression are actually dominated by a more expensive backpropagation-based step. It remains open if similar results can be obtained by a fully tuning-free algorithm.

Q4. *Why do you use matrix factorization, and SVD in particular?*

A4. Committing to a backpropagation-based algorithm (see Q3) means that we have to deal with increased memory requirements. As such, matrix factorization is not helpful in that respect because the number of parameters might even increase initially (for instance, an SVD-parametrization basically doubles the size of a quadratic weight matrix). On the other hand, tuning and pruning only the bottleneck layer (i.e., the singular value masks in case of `ACIP`) has the potential for drastic size reductions and is highly parameter-efficient. For example, the number of tunable mask parameters for LLaMA-7B with `ACIP` is <1M.

With this in mind, SVD as a specific matrix factorization is an obvious candidate due to its beneficial mathematical and numerical properties, in particular, optimal low-rank matrix approximation and stable matrix operations due to orthogonality.

## B   Implementation Details

In this section, we report more technical details and hyperparameters used for our experiments.

**Dataset and models.**   Following previous work on LLM compression, we use C4 (Raffel et al., 2019) for training as it is a good proxy of a general-purpose dataset. In the context of `ACIP`, it should be primarily seen as a calibration dataset that allows us to propagate meaningful activations through a pre-trained model while performing structured pruning. Overfitting to the distribution of C4 is implicitly mitigated, since we only tune very few parameters (masks and low-rank adapters) compared to the total model size. As loss function $\mathcal{L}$ in (8), we use the standard negative log-likelihood loss for next-token prediction.

All considered (evaluation) datasets and pre-trained models are imported with the HuggingFace `transformers`-library in `bfloat16`-precision. Our experiments were implemented with `PyTorch` (Paszke et al., 2019) and the `Lightning` package.

**ACIP-specifics.**   As mentioned in Remark 2.1, we apply a linear scheduler that increases the regularization parameter $\lambda$ dynamically over the pruning process. This ensures that the pruning becomes more and more aggressive over time and the stopping criterion will be reached at some point. Across all experiments, we use $\lambda = 1e{-}3$ as initial value and increase it by a factor of 1.01 every 4 steps (this amounts to a doubling of $\lambda$ at about every 280 steps).

As pointed out in Section 2.2.2, we choose a target compression rate as a stopping criterion for `ACIP`. In most experiments, a rate of $r_{\text{stop}} = 0.4$ is reasonable (i.e., only 40% or the original parameters remain), and we refer to Appendix D.4 for further discussion and analysis. After the stopping criterion is reached, we tune the low-rank adapter for 1k more steps while the masks are frozen (see Section 2.2.2).

The mask parameters in (7) are rescaled by a fixed factor of 0.02 to ensure a better alignment with the numerical range of the remaining network weights. The low-rank adapters are created with $r = 32$, $\alpha = 16$, and dropout 0.05. For LLaMA-7B, the number of tunable parameters amounts to $<$1M mask parameters and approximately 80M low-rank adapter parameters.

For sample data from C4, we use 1024 tokens per sample and a batch size of 4. We use Adam (Kingma & Ba, 2015) as optimizer without weight decay and a learning rate of $5e{-}5$.

**Runtime analysis.**   `ACIP` requires significantly fewer steps than fine-tuning. Depending on when the stopping criterion is reached, it typically takes 1.5k - 2.5k steps, including 1k post-tuning steps of the low-rank adapters. For LLaMA-7B, for example, this amounts to a wall clock runtime of $<$ 30 minutes, including the initial SVD computations for the base model parametrization. All runs were performed on single NVIDIA H100 GPUs. See also Appendix D.1 for a more detailed efficiency analysis.

**Fine-tuning.**   In all post-compression fine-tunings (see Section 3.3), we simply continue training `ACIP`'s low-rank adapters (the optimizer states are reset). We train for 25k steps on C4 with a batch size of 4 and a learning rate of $2e{-}4$.

## C    Supplementary Results for Section 3.1 – Section 3.4

Figure A8 complements the trade-off curves in Figures 4 and 5 by all other considered evaluation metrics (see Section 3.1). Table A2 reports these results in terms of numbers, including all fine-tuning results for all models (see Section 3.3). Table A3 analyzes the effect of fine-tuning of `ACIP` compared to existing SVD-based compression methods. Table A4 provides more detailed evaluation results on fine-tuning a quantized and compressed LLaMA-7B model (see Section 3.4).

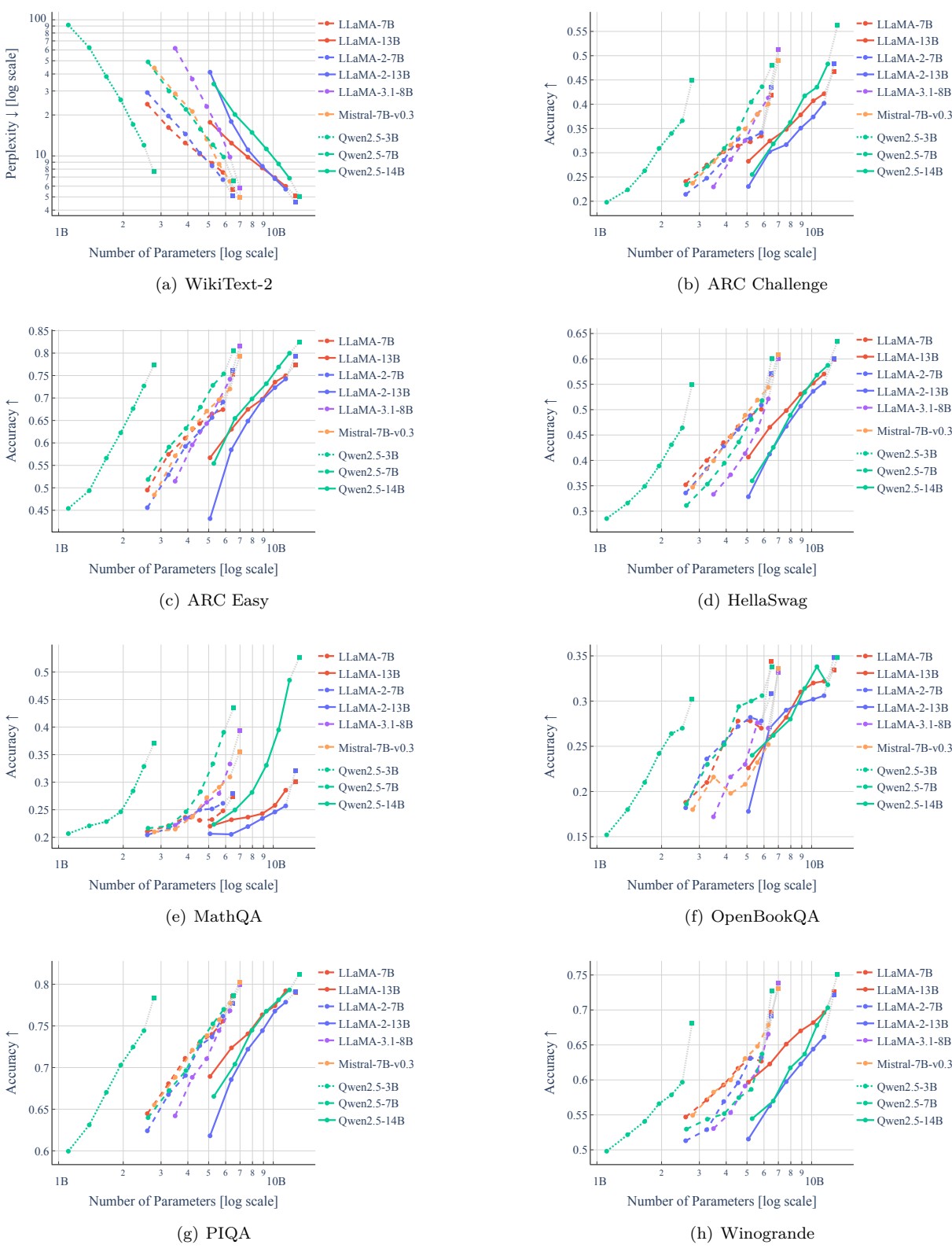

Figure A8: Compression-performance trade-off curves generated by `ACIP` on **WikiText-2** and **individual LM-Eval tasks**, complementing the results of Figures 4 and 5.

Table A2: Evaluation results for `ACIP` on all considered LLMs. Scores on C4 and WikiText-2 are measured in perplexity (smaller is better), and the LM-Eval zero-shot tasks are measured in accuracy (higher is better). *The results of LLaMA2-13B were achieved by ignoring all up-projection layers in `ACIP` (see Appendix D.8).

| Model | Type Size | C4 ↓ ACIP | C4 ↓ FT | WikiText-2 ↓ ACIP | WikiText-2 ↓ FT | ARC_c ↑ ACIP | ARC_c ↑ FT | ARC_e ↑ ACIP | ARC_e ↑ FT | HellaS. ↑ ACIP | HellaS. ↑ FT | MathQA ↑ ACIP | MathQA ↑ FT | Openb. ↑ ACIP | Openb. ↑ FT | PIQA ↑ ACIP | PIQA ↑ FT | WinoG. ↑ ACIP | WinoG. ↑ FT | LM Eval Avg. ↑ ACIP | LM Eval Avg. ↑ FT |
|---|---|---|---|---|---|---|---|---|---|---|---|---|---|---|---|---|---|---|---|---|---|
| LLaMA-7B | 40% | 21.05 | 15.66 | 23.99 | 17.33 | 0.24 | 0.24 | 0.49 | 0.53 | 0.35 | 0.38 | 0.21 | 0.21 | 0.19 | 0.20 | 0.64 | 0.67 | 0.55 | 0.57 | 0.38 | 0.40 |
| | 50% | 16.47 | 12.89 | 16.16 | 11.88 | 0.27 | 0.27 | 0.57 | 0.59 | 0.40 | 0.43 | 0.22 | 0.23 | 0.21 | 0.23 | 0.68 | 0.71 | 0.57 | 0.60 | 0.42 | 0.44 |
| | 60% | 13.91 | 11.14 | 12.46 | 9.63 | 0.30 | 0.32 | 0.61 | 0.64 | 0.44 | 0.47 | 0.24 | 0.23 | 0.25 | 0.26 | 0.71 | 0.73 | 0.59 | 0.63 | 0.45 | 0.47 |
| | 70% | 12.22 | 9.84 | 10.35 | 8.27 | 0.31 | 0.34 | 0.64 | 0.69 | 0.47 | 0.50 | 0.23 | 0.24 | 0.28 | 0.28 | 0.73 | 0.75 | 0.62 | 0.66 | 0.47 | 0.49 |
| | 80% | 10.92 | 8.81 | 8.83 | 7.19 | 0.32 | 0.38 | 0.66 | 0.71 | 0.49 | 0.53 | 0.23 | 0.24 | 0.28 | 0.31 | 0.74 | 0.77 | 0.63 | 0.68 | 0.48 | 0.52 |
| | 90% | 9.75 | 7.69 | 7.56 | 6.12 | 0.33 | 0.39 | 0.67 | 0.73 | 0.50 | 0.56 | 0.25 | 0.25 | 0.27 | 0.33 | 0.76 | 0.78 | 0.63 | 0.69 | 0.49 | 0.53 |
| | 100% | 9.52 | 7.32 | 7.20 | 5.75 | 0.33 | 0.40 | 0.69 | 0.75 | 0.51 | 0.57 | 0.25 | 0.26 | 0.26 | 0.34 | 0.76 | 0.79 | 0.63 | 0.70 | 0.49 | 0.54 |
| | Orig. | 7.31 | | 5.68 | | 0.42 | | 0.75 | | 0.57 | | 0.27 | | 0.34 | | 0.79 | | 0.70 | | 0.55 | |
| LLaMA-13B | 40% | 16.64 | 13.38 | 17.66 | 13.42 | 0.28 | 0.28 | 0.57 | 0.59 | 0.41 | 0.43 | 0.22 | 0.23 | 0.23 | 0.24 | 0.69 | 0.70 | 0.60 | 0.62 | 0.43 | 0.44 |
| | 50% | 13.06 | 11.12 | 12.42 | 10.49 | 0.32 | 0.34 | 0.63 | 0.63 | 0.47 | 0.48 | 0.23 | 0.23 | 0.26 | 0.28 | 0.72 | 0.73 | 0.62 | 0.64 | 0.47 | 0.47 |
| | 60% | 11.33 | 9.76 | 9.79 | 8.30 | 0.35 | 0.33 | 0.67 | 0.68 | 0.50 | 0.51 | 0.24 | 0.24 | 0.28 | 0.30 | 0.74 | 0.76 | 0.65 | 0.67 | 0.49 | 0.50 |
| | 70% | 10.10 | 8.72 | 8.17 | 7.06 | 0.38 | 0.38 | 0.70 | 0.69 | 0.53 | 0.54 | 0.24 | 0.26 | 0.31 | 0.31 | 0.76 | 0.77 | 0.67 | 0.70 | 0.51 | 0.52 |
| | 80% | 9.06 | 7.91 | 6.91 | 6.21 | 0.41 | 0.41 | 0.74 | 0.74 | 0.55 | 0.57 | 0.26 | 0.27 | 0.32 | 0.34 | 0.77 | 0.78 | 0.68 | 0.69 | 0.53 | 0.54 |
| | 90% | 8.04 | 7.06 | 5.98 | 5.40 | 0.42 | 0.44 | 0.75 | 0.76 | 0.57 | 0.59 | 0.29 | 0.29 | 0.32 | 0.33 | 0.79 | 0.79 | 0.70 | 0.72 | 0.55 | 0.56 |
| | 100% | 7.86 | 6.79 | 5.83 | 5.15 | 0.42 | 0.46 | 0.75 | 0.77 | 0.57 | 0.60 | 0.29 | 0.30 | 0.31 | 0.32 | 0.79 | 0.79 | 0.70 | 0.73 | 0.55 | 0.57 |
| | Orig. | 6.77 | | 5.09 | | 0.47 | | 0.77 | | 0.60 | | 0.30 | | 0.33 | | 0.79 | | 0.73 | | 0.57 | |
| LLaMA-2-7B | 40% | 24.62 | 16.74 | 29.20 | 18.00 | 0.21 | 0.22 | 0.46 | 0.50 | 0.34 | 0.37 | 0.20 | 0.21 | 0.18 | 0.19 | 0.62 | 0.66 | 0.51 | 0.55 | 0.36 | 0.39 |
| | 50% | 18.36 | 13.32 | 19.64 | 12.25 | 0.25 | 0.27 | 0.53 | 0.57 | 0.38 | 0.42 | 0.22 | 0.22 | 0.24 | 0.23 | 0.67 | 0.69 | 0.53 | 0.56 | 0.40 | 0.42 |
| | 60% | 15.27 | 11.15 | 14.47 | 9.54 | 0.28 | 0.31 | 0.59 | 0.63 | 0.43 | 0.46 | 0.23 | 0.24 | 0.25 | 0.23 | 0.69 | 0.72 | 0.57 | 0.62 | 0.44 | 0.46 |
| | 70% | 12.96 | 9.73 | 10.47 | 7.74 | 0.33 | 0.35 | 0.62 | 0.68 | 0.46 | 0.50 | 0.25 | 0.24 | 0.27 | 0.26 | 0.73 | 0.74 | 0.60 | 0.64 | 0.47 | 0.49 |
| | 80% | 11.31 | 8.63 | 8.46 | 6.54 | 0.33 | 0.37 | 0.66 | 0.70 | 0.49 | 0.53 | 0.25 | 0.26 | 0.28 | 0.31 | 0.74 | 0.77 | 0.63 | 0.66 | 0.48 | 0.51 |
| | 90% | 9.46 | 7.43 | 6.69 | 5.45 | 0.34 | 0.43 | 0.69 | 0.75 | 0.51 | 0.56 | 0.26 | 0.28 | 0.28 | 0.33 | 0.76 | 0.78 | 0.63 | 0.69 | 0.50 | 0.54 |
| | 100% | 9.34 | 7.06 | 6.54 | 5.13 | 0.34 | 0.43 | 0.70 | 0.76 | 0.51 | 0.57 | 0.26 | 0.28 | 0.27 | 0.32 | 0.76 | 0.78 | 0.64 | 0.69 | 0.50 | 0.55 |
| | Orig. | 7.04 | | 5.11 | | 0.44 | | 0.76 | | 0.57 | | 0.28 | | 0.31 | | 0.78 | | 0.69 | | 0.55 | |
| LLaMA-2-13B | 40% | 27.55 | 84.28 | 41.22 | 145.79 | 0.23 | 0.23 | 0.43 | 0.46 | 0.33 | 0.30 | 0.21 | 0.21 | 0.18 | 0.18 | 0.62 | 0.63 | 0.52 | 0.52 | 0.36 | 0.36 |
| | 50% | 17.10 | 12.76 | 17.89 | 13.12 | 0.30 | 0.32 | 0.58 | 0.61 | 0.41 | 0.44 | 0.21 | 0.23 | 0.27 | 0.27 | 0.69 | 0.70 | 0.56 | 0.58 | 0.43 | 0.45 |
| | 60% | 13.29 | 10.05 | 11.11 | 8.43 | 0.32 | 0.36 | 0.65 | 0.68 | 0.47 | 0.50 | 0.22 | 0.23 | 0.29 | 0.31 | 0.72 | 0.74 | 0.60 | 0.63 | 0.47 | 0.49 |
| | 70% | 11.04 | 8.64 | 8.40 | 6.71 | 0.35 | 0.41 | 0.70 | 0.74 | 0.51 | 0.54 | 0.23 | 0.25 | 0.30 | 0.33 | 0.74 | 0.77 | 0.62 | 0.66 | 0.49 | 0.53 |
| | 80% | 9.54 | 7.68 | 6.80 | 5.66 | 0.37 | 0.44 | 0.72 | 0.76 | 0.54 | 0.57 | 0.25 | 0.28 | 0.30 | 0.34 | 0.77 | 0.78 | 0.64 | 0.71 | 0.51 | 0.55 |
| | 90% | 8.26 | 6.86 | 5.70 | 4.87 | 0.40 | 0.47 | 0.74 | 0.78 | 0.55 | 0.60 | 0.26 | 0.31 | 0.31 | 0.34 | 0.78 | 0.79 | 0.66 | 0.72 | 0.53 | 0.57 |
| | 100% | 7.87 | 6.56 | 5.42 | 4.61 | 0.41 | 0.47 | 0.75 | 0.79 | 0.56 | 0.60 | 0.28 | 0.32 | 0.31 | 0.35 | 0.78 | 0.79 | 0.69 | 0.71 | 0.54 | 0.58 |
| | Orig. | 6.52 | | 4.57 | | 0.48 | | 0.79 | | 0.60 | | 0.32 | | 0.35 | | 0.79 | | 0.72 | | 0.58 | |
| LLaMA-3.1-8B | 50% | 43.32 | 26.52 | 61.77 | 29.52 | 0.23 | 0.27 | 0.51 | 0.58 | 0.33 | 0.37 | 0.22 | 0.23 | 0.17 | 0.19 | 0.64 | 0.68 | 0.53 | 0.54 | 0.38 | 0.41 |
| | 60% | 31.55 | 21.00 | 36.69 | 19.26 | 0.29 | 0.29 | 0.60 | 0.61 | 0.37 | 0.42 | 0.24 | 0.24 | 0.22 | 0.23 | 0.69 | 0.72 | 0.55 | 0.56 | 0.42 | 0.44 |
| | 70% | 24.90 | 17.08 | 23.06 | 13.55 | 0.33 | 0.32 | 0.64 | 0.66 | 0.41 | 0.47 | 0.26 | 0.27 | 0.23 | 0.27 | 0.71 | 0.73 | 0.59 | 0.62 | 0.45 | 0.48 |
| | 80% | 20.78 | 14.21 | 15.60 | 10.12 | 0.38 | 0.40 | 0.69 | 0.72 | 0.46 | 0.51 | 0.28 | 0.30 | 0.28 | 0.29 | 0.74 | 0.77 | 0.61 | 0.66 | 0.49 | 0.52 |
| | 90% | 16.25 | 11.28 | 9.80 | 7.24 | 0.41 | 0.48 | 0.74 | 0.80 | 0.52 | 0.57 | 0.33 | 0.35 | 0.27 | 0.31 | 0.77 | 0.79 | 0.67 | 0.71 | 0.53 | 0.57 |
| | 100% | 14.57 | 9.42 | 8.04 | 5.95 | 0.40 | 0.51 | 0.75 | 0.82 | 0.53 | 0.60 | 0.36 | 0.40 | 0.27 | 0.34 | 0.78 | 0.80 | 0.66 | 0.74 | 0.54 | 0.60 |
| | Orig. | 9.31 | | 5.86 | | 0.51 | | 0.82 | | 0.60 | | 0.39 | | 0.33 | | 0.80 | | 0.74 | | 0.60 | |
| Mistral-7B-v0.3 | 40% | 28.92 | 19.21 | 44.29 | 23.24 | 0.24 | 0.26 | 0.48 | 0.53 | 0.35 | 0.38 | 0.21 | 0.21 | 0.18 | 0.19 | 0.66 | 0.67 | 0.55 | 0.57 | 0.38 | 0.40 |
| | 50% | 21.44 | 14.86 | 28.60 | 16.53 | 0.28 | 0.28 | 0.57 | 0.59 | 0.40 | 0.43 | 0.21 | 0.23 | 0.22 | 0.21 | 0.69 | 0.70 | 0.58 | 0.60 | 0.42 | 0.43 |
| | 60% | 16.89 | 12.49 | 21.19 | 12.29 | 0.32 | 0.32 | 0.63 | 0.66 | 0.45 | 0.48 | 0.24 | 0.24 | 0.20 | 0.23 | 0.72 | 0.73 | 0.60 | 0.62 | 0.45 | 0.47 |
| | 70% | 13.75 | 10.95 | 13.28 | 9.69 | 0.35 | 0.34 | 0.67 | 0.68 | 0.49 | 0.52 | 0.27 | 0.28 | 0.21 | 0.26 | 0.74 | 0.76 | 0.63 | 0.63 | 0.48 | 0.49 |
| | 80% | 11.80 | 9.84 | 8.70 | 7.49 | 0.38 | 0.39 | 0.70 | 0.73 | 0.52 | 0.55 | 0.29 | 0.29 | 0.23 | 0.26 | 0.76 | 0.77 | 0.65 | 0.69 | 0.50 | 0.53 |
| | 90% | 10.42 | 8.84 | 6.51 | 5.85 | 0.40 | 0.43 | 0.72 | 0.75 | 0.54 | 0.59 | 0.31 | 0.33 | 0.25 | 0.31 | 0.78 | 0.79 | 0.68 | 0.70 | 0.53 | 0.56 |
| | 100% | 9.85 | 8.31 | 6.04 | 5.31 | 0.40 | 0.45 | 0.73 | 0.77 | 0.55 | 0.60 | 0.33 | 0.34 | 0.26 | 0.33 | 0.78 | 0.79 | 0.68 | 0.72 | 0.53 | 0.57 |
| | Orig. | 8.05 | | 4.96 | | 0.49 | | 0.79 | | 0.61 | | 0.36 | | 0.34 | | 0.80 | | 0.73 | | 0.59 | |
| Qwen2.5-3B | 40% | 71.23 | 36.85 | 91.51 | 39.44 | 0.20 | 0.22 | 0.45 | 0.51 | 0.29 | 0.31 | 0.21 | 0.22 | 0.15 | 0.16 | 0.60 | 0.64 | 0.50 | 0.52 | 0.34 | 0.37 |
| | 50% | 57.17 | 29.43 | 62.42 | 26.92 | 0.22 | 0.25 | 0.49 | 0.57 | 0.32 | 0.34 | 0.22 | 0.21 | 0.18 | 0.19 | 0.63 | 0.67 | 0.52 | 0.53 | 0.37 | 0.39 |
| | 60% | 43.30 | 23.30 | 38.26 | 18.38 | 0.26 | 0.29 | 0.57 | 0.63 | 0.35 | 0.39 | 0.23 | 0.23 | 0.21 | 0.24 | 0.67 | 0.69 | 0.54 | 0.56 | 0.40 | 0.43 |
| | 70% | 34.24 | 19.04 | 25.81 | 13.50 | 0.31 | 0.34 | 0.62 | 0.68 | 0.39 | 0.43 | 0.25 | 0.26 | 0.24 | 0.26 | 0.70 | 0.72 | 0.57 | 0.58 | 0.44 | 0.47 |
| | 80% | 25.50 | 16.16 | 17.02 | 10.68 | 0.34 | 0.36 | 0.68 | 0.73 | 0.43 | 0.47 | 0.28 | 0.31 | 0.26 | 0.24 | 0.72 | 0.74 | 0.58 | 0.57 | 0.47 | 0.49 |
| | 90% | 20.10 | 13.82 | 11.99 | 8.57 | 0.37 | 0.42 | 0.73 | 0.77 | 0.46 | 0.52 | 0.33 | 0.36 | 0.27 | 0.33 | 0.74 | 0.78 | 0.60 | 0.67 | 0.50 | 0.55 |
| | 100% | 18.73 | 12.81 | 10.63 | 7.70 | 0.36 | 0.47 | 0.72 | 0.78 | 0.46 | 0.54 | 0.33 | 0.41 | 0.24 | 0.31 | 0.74 | 0.78 | 0.60 | 0.69 | 0.49 | 0.57 |
| | Orig. | 12.90 | | 7.64 | | 0.45 | | 0.77 | | 0.55 | | 0.37 | | 0.30 | | 0.78 | | 0.68 | | 0.56 | |
| Qwen2.5-7B | 40% | 46.43 | 29.26 | 49.04 | 27.24 | 0.23 | 0.24 | 0.52 | 0.58 | 0.31 | 0.34 | 0.22 | 0.21 | 0.19 | 0.20 | 0.64 | 0.67 | 0.53 | 0.53 | 0.38 | 0.40 |
| | 50% | 34.90 | 23.26 | 29.96 | 19.72 | 0.27 | 0.30 | 0.59 | 0.64 | 0.35 | 0.39 | 0.22 | 0.23 | 0.23 | 0.25 | 0.67 | 0.70 | 0.54 | 0.57 | 0.41 | 0.44 |
| | 60% | 27.84 | 18.73 | 21.98 | 13.73 | 0.31 | 0.33 | 0.63 | 0.68 | 0.39 | 0.45 | 0.25 | 0.26 | 0.25 | 0.28 | 0.70 | 0.73 | 0.55 | 0.60 | 0.44 | 0.48 |
| | 70% | 22.97 | 15.96 | 15.72 | 10.71 | 0.35 | 0.42 | 0.68 | 0.74 | 0.44 | 0.49 | 0.28 | 0.30 | 0.29 | 0.30 | 0.73 | 0.76 | 0.57 | 0.63 | 0.48 | 0.52 |
| | 80% | 19.68 | 14.02 | 12.07 | 8.89 | 0.40 | 0.46 | 0.73 | 0.78 | 0.48 | 0.53 | 0.33 | 0.36 | 0.30 | 0.33 | 0.75 | 0.78 | 0.59 | 0.67 | 0.51 | 0.56 |
| | 90% | 17.09 | 12.59 | 9.82 | 7.63 | 0.44 | 0.48 | 0.75 | 0.80 | 0.52 | 0.56 | 0.39 | 0.41 | 0.31 | 0.32 | 0.77 | 0.79 | 0.64 | 0.70 | 0.54 | 0.58 |
| | 100% | 15.34 | 11.43 | 8.38 | 6.60 | 0.43 | 0.50 | 0.75 | 0.82 | 0.53 | 0.59 | 0.43 | 0.46 | 0.28 | 0.34 | 0.77 | 0.79 | 0.66 | 0.72 | 0.55 | 0.60 |
| | Orig. | 11.47 | | 6.55 | | 0.48 | | 0.81 | | 0.60 | | 0.43 | | 0.34 | | 0.79 | | 0.73 | | 0.60 | |
| Qwen2.5-14B | 40% | 36.51 | 25.58 | 33.78 | 22.22 | 0.26 | 0.29 | 0.55 | 0.61 | 0.36 | 0.38 | 0.22 | 0.23 | 0.24 | 0.26 | 0.67 | 0.68 | 0.54 | 0.57 | 0.41 | 0.43 |
| | 50% | 26.27 | 19.53 | 20.15 | 14.57 | 0.32 | 0.33 | 0.65 | 0.68 | 0.43 | 0.44 | 0.25 | 0.25 | 0.26 | 0.27 | 0.70 | 0.71 | 0.57 | 0.59 | 0.45 | 0.47 |
| | 60% | 21.29 | 15.63 | 14.87 | 10.48 | 0.36 | 0.40 | 0.70 | 0.73 | 0.49 | 0.51 | 0.28 | 0.32 | 0.28 | 0.31 | 0.74 | 0.76 | 0.62 | 0.66 | 0.50 | 0.53 |
| | 70% | 17.99 | 13.99 | 11.25 | 8.89 | 0.42 | 0.43 | 0.73 | 0.76 | 0.53 | 0.54 | 0.33 | 0.36 | 0.31 | 0.32 | 0.77 | 0.78 | 0.64 | 0.68 | 0.53 | 0.55 |
| | 80% | 15.23 | 11.97 | 8.73 | 7.11 | 0.44 | 0.48 | 0.77 | 0.78 | 0.57 | 0.58 | 0.39 | 0.44 | 0.34 | 0.36 | 0.78 | 0.79 | 0.68 | 0.73 | 0.57 | 0.60 |
| | 90% | 13.05 | 10.77 | 6.86 | 6.04 | 0.48 | 0.52 | 0.80 | 0.82 | 0.59 | 0.60 | 0.49 | 0.49 | 0.32 | 0.35 | 0.79 | 0.81 | 0.70 | 0.74 | 0.60 | 0.62 |
| | 100% | 12.37 | 9.98 | 6.23 | 5.11 | 0.47 | 0.53 | 0.79 | 0.82 | 0.58 | 0.62 | 0.51 | 0.53 | 0.32 | 0.35 | 0.79 | 0.81 | 0.73 | 0.77 | 0.60 | 0.63 |
| | Orig. | 9.99 | | 5.05 | | 0.56 | | 0.82 | | 0.63 | | 0.53 | | 0.35 | | 0.81 | | 0.75 | | 0.64 | |

Table A3: Evaluation of **LLaMA-7B** on **WikiText-2** (perplexity, smaller is better) under different compression ratios, with and without post-training fine-tuning. We compare `ACIP` with the existing SVD-based compression methods ASVD (Yuan et al., 2024) and SVD-LLM (Wang et al., 2024), see also Section 3.2. The scores for ASVD and SVD-LLM are taken from Wang et al. (2024, Table 4). Note that `ACIP` was fine-tuned on C4, while ASVD and SVD-LLM fine-tuned on WikiText-2 directly.

| Compression Ratio
Method | 40% | 50% | 60% | 70% | 80% |
|---|---|---|---|---|---|
| ASVD | 57057.00 | 15358.00 | 1407.00 | 51.00 | 11.14 |
| ASVD + LoRA FT | 44.81 | 21.83 | 14.86 | 10.16 | 8.37 |
| SVD-LLM | 42.30 | 23.97 | 13.11 | 9.56 | 7.94 |
| SVD-LLM + LoRA FT | 17.93 | 13.26 | 10.65 | 9.14 | 7.78 |
| `ACIP` | 24.00 | 16.17 | 12.46 | 10.34 | 8.83 |
| `ACIP` + FT | 17.33 | 11.88 | 9.63 | 8.27 | 7.19 |

Table A4: More detailed evaluation results for our quantization experiments in Section 3.4, reported in terms of numbers.

| Size | Ablation | Eff. model size [GB] | C4 ↓ | WikiText-2 ↓ | ARC_c ↑ | ARC_e ↑ | HellaS. ↑ | MathQA ↑ | Openb. ↑ | PIQA ↑ | WinoG. ↑ | LM Eval Avg. ↑ |
|---|---|---|---|---|---|---|---|---|---|---|---|---|
| 40% | ACIP | 5.47 | 21.05 | 24.00 | 0.24 | 0.49 | 0.35 | 0.21 | 0.19 | 0.65 | 0.55 | 0.38 |
| | ACIP → FT | 5.47 | 15.66 | 17.33 | 0.24 | 0.53 | 0.38 | 0.21 | 0.20 | 0.67 | 0.57 | 0.40 |
| | ACIP → W4A16 | 1.89 | 27.12 | 35.40 | 0.22 | 0.46 | 0.33 | 0.20 | 0.18 | 0.62 | 0.53 | 0.36 |
| | ACIP → W4A16 → FT | 1.89 | 16.67 | 18.90 | 0.23 | 0.52 | 0.37 | 0.22 | 0.20 | 0.67 | 0.57 | 0.40 |
| 50% | ACIP | 6.70 | 16.47 | 16.17 | 0.28 | 0.58 | 0.40 | 0.22 | 0.21 | 0.68 | 0.57 | 0.42 |
| | ACIP → FT | 6.70 | 12.89 | 11.88 | 0.27 | 0.59 | 0.43 | 0.23 | 0.23 | 0.71 | 0.60 | 0.44 |
| | ACIP → W4A16 | 2.21 | 19.28 | 19.96 | 0.26 | 0.54 | 0.37 | 0.21 | 0.20 | 0.67 | 0.55 | 0.40 |
| | ACIP → W4A16 → FT | 2.21 | 13.85 | 13.33 | 0.25 | 0.58 | 0.41 | 0.23 | 0.23 | 0.70 | 0.59 | 0.43 |
| 60% | ACIP | 7.88 | 13.91 | 12.46 | 0.30 | 0.61 | 0.43 | 0.23 | 0.25 | 0.71 | 0.60 | 0.45 |
| | ACIP → FT | 7.88 | 11.14 | 9.63 | 0.32 | 0.64 | 0.47 | 0.23 | 0.26 | 0.73 | 0.63 | 0.47 |
| | ACIP → W4A16 | 2.51 | 15.84 | 14.64 | 0.29 | 0.58 | 0.42 | 0.22 | 0.22 | 0.69 | 0.57 | 0.43 |
| | ACIP → W4A16 → FT | 2.51 | 11.77 | 10.31 | 0.29 | 0.64 | 0.45 | 0.22 | 0.26 | 0.72 | 0.61 | 0.46 |
| 70% | ACIP | 9.10 | 12.22 | 10.34 | 0.31 | 0.64 | 0.47 | 0.23 | 0.27 | 0.73 | 0.62 | 0.47 |
| | ACIP → FT | 9.10 | 9.84 | 8.27 | 0.34 | 0.69 | 0.50 | 0.24 | 0.28 | 0.75 | 0.66 | 0.49 |
| | ACIP → W4A16 | 2.83 | 13.45 | 11.80 | 0.29 | 0.63 | 0.45 | 0.23 | 0.24 | 0.72 | 0.60 | 0.45 |
| | ACIP → W4A16 → FT | 2.83 | 10.38 | 8.74 | 0.32 | 0.67 | 0.48 | 0.23 | 0.28 | 0.75 | 0.64 | 0.48 |
| 80% | ACIP | 10.30 | 10.91 | 8.83 | 0.33 | 0.67 | 0.49 | 0.23 | 0.28 | 0.74 | 0.63 | 0.48 |
| | ACIP → FT | 10.30 | 8.81 | 7.19 | 0.38 | 0.71 | 0.53 | 0.24 | 0.31 | 0.77 | 0.68 | 0.52 |
| | ACIP → W4A16 | 3.13 | 12.61 | 9.87 | 0.32 | 0.65 | 0.47 | 0.23 | 0.28 | 0.74 | 0.61 | 0.47 |
| | ACIP → W4A16 → FT | 3.13 | 9.26 | 7.60 | 0.36 | 0.69 | 0.52 | 0.24 | 0.30 | 0.76 | 0.66 | 0.50 |
| 90% | ACIP | 11.50 | 9.75 | 7.56 | 0.34 | 0.68 | 0.50 | 0.25 | 0.27 | 0.75 | 0.63 | 0.49 |
| | ACIP → FT | 11.50 | 7.69 | 6.12 | 0.39 | 0.73 | 0.56 | 0.25 | 0.33 | 0.78 | 0.69 | 0.53 |
| | ACIP → W4A16 | 3.44 | 10.25 | 7.90 | 0.32 | 0.66 | 0.50 | 0.24 | 0.27 | 0.75 | 0.63 | 0.48 |
| | ACIP → W4A16 → FT | 3.44 | 7.97 | 6.39 | 0.38 | 0.73 | 0.56 | 0.25 | 0.35 | 0.78 | 0.69 | 0.53 |
| 100% | ACIP | 12.70 | 9.52 | 7.20 | 0.33 | 0.69 | 0.51 | 0.25 | 0.26 | 0.76 | 0.63 | 0.49 |
| | ACIP → FT | 12.70 | 7.32 | 5.75 | 0.40 | 0.75 | 0.57 | 0.26 | 0.34 | 0.79 | 0.70 | 0.54 |
| | ACIP → W4A16 | 3.75 | 9.75 | 7.37 | 0.34 | 0.69 | 0.51 | 0.25 | 0.27 | 0.76 | 0.63 | 0.49 |
| | ACIP → W4A16 → FT | 3.75 | 7.52 | 5.94 | 0.40 | 0.75 | 0.56 | 0.27 | 0.34 | 0.78 | 0.69 | 0.54 |

# D   Further Experiments and Ablations

In this section, we present several supplementary experiments analyzing the impact of some key algorithmic components and design choices of `ACIP` in more detail. Note that the most detailed analyses and ablations are carried out with LLaMA-7B as it was most extensively studied in previous research on structured weight pruning.

## D.1   Efficiency Analysis

Table A5 reports several statistics on the efficiency of the `ACIP` algorithm and inference speed of compressed models. While these preliminary results do not immediately indicate gains in inference speed, we expect that further optimization like merging the low-rank adapters can compensate for the matrix-factorization overhead (one additional matrix-vector multiplication) and outperform the base model. Moreover, we note that compared to performance-size trade-offs, which are our main concern, analyzing inference speed-ups requires a very careful consideration about the hardware in use (accelerator model, parallel processing units, etc.) and measurement setup (sequence length, batch size, etc.).

Table A5: Efficiency analysis of `ACIP` for **LLaMA-7B**. The first three rows report the runtime and memory statistics of `ACIP`'s key steps (see Section 2.2 and Figure 2) both in terms of numbers and their qualitative asymptotics. Here, the model sizes are measured as (uncompressed) checkpoint sizes. "Runtime pruning" refers to the process of pruning the mask parameters to a desired compression ratio (revertible), whereas "Runtime compress" refers to the process of discarding pruned singular vectors and possibly unparametrizing linear layers, so that the model gets actually compressed (see Step 3 in Section 2.2.3). The statistics of inference speed were obtained by generating new text of sequence length 64 and batch size 64. To measure FLOPs, we use the `fvcore` package and an input sequence of length 512.

| Stage | Metric | LLaMA-7B |
|---|---|---:|
| ACIP Step 1 (Model Reparametrization) $\mathcal{O}(\#\text{Layers} \times \text{SVD of Layer})$ | Runtime [min] | 4.95 |
| | Size parametrized model [GB] | 19.71 |
| | Size base model [GB] | 12.70 |
| ACIP Step 2 (Scoring by Iterative Pruning) $\mathcal{O}(\#\text{Steps of Masks \& LoRA Updates})$ | Runtime [min] | 23.12 |
| | Reserved GPU memory peak [GB] | 62.45 |
| | Steps / s | 1.68 |
| ACIP Step 3 (Any Compression) $\mathcal{O}(\#\text{Layers} \times \text{Layer Input Dimension})$ | Runtime pruning [s] | 0.49 |
| | Runtime compress [s] | 0.18 |
| Inference at 40% Size | Size model [GB] | 5.47 |
| | Reserved GPU memory peak [GB] | 25.68 |
| | Latency [s] | 2.57 |
| | Tokens / s | 1594.99 |
| | GigaFLOPs | 1335.85 |
| Inference at 70% Size | Size model [GB] | 9.10 |
| | Reserved GPU memory peak [GB] | 29.43 |
| | Latency [s] | 2.47 |
| | Tokens / s | 1658.63 |
| | GigaFLOPs | 2265.03 |
| Inference at 100% Size (Original) | Size model [GB] | 12.70 |
| | Reserved GPU memory peak [GB] | 32.79 |
| | Latency [s] | 1.67 |
| | Tokens / s | 2447.75 |
| | GigaFLOPs | 3188.63 |

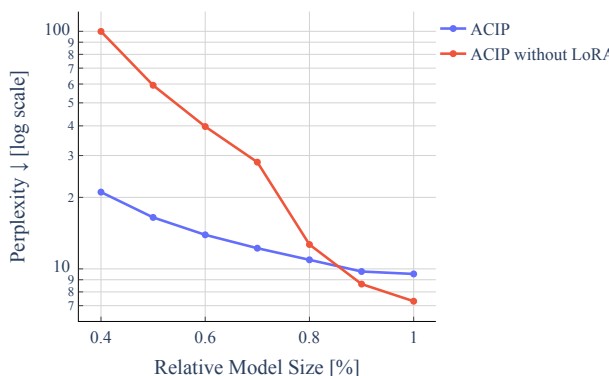

Figure A9: Compression-performance trade-off curves for **LLaMA-7B** on **C4** with and without using a LoRA-adapter for correction in `ACIP`.

### D.2 Impact of Low-Rank Adapters

The primary purpose of the low-rank adapters used in `ACIP` is to correct compression errors on-the-fly during the optimization. A surprising finding of our work is that the final adapters are "universal" in the sense that they can be used across all seen compression levels. While we expect that other PEFT-style approaches would lead to similar findings, it is natural to ask how `ACIP` would perform without any correction, i.e., just the mask parameters are tuned according to (8). This ablation study is shown in Figure A9. While performing significantly worse than with LoRA, we observe that the perplexity does not blow up and the results are even slightly better than SVD-LLM (see Table 1). This stable behavior of `ACIP` is closely related to our parameterization of the mask in (7) which ensures that the forward pass corresponds to the actual outputs of the pruned model with binary masks. On the other hand, the straight-through estimator still enables backpropagation.

### D.3 Impact of the Compression Rule

The Any Compression stage of `ACIP` (Step 3 in Section 2.2.3) relies on a simple algorithmic rule: Prune as many singular values according to the learned score map $\rho$ as needed for a desired compression rate $r$. Here, the tuned low-rank adapters $\mathbf{\Delta}$ are used across all compression levels, even for $r = 1.0$, i.e., the size of the (uncompressed) base model. While the usage of low-rank adapters is helpful for simultaneous error correction along with iterative pruning (Step 2 in Section 2.2.2), it can lead to a slight drop in performance for lower compression levels. This became already apparent in the ablation of Figure A9 above, where LoRA was fully omitted.

Figure A10 shows that this performance gap can be reduced by a refinement of Step 3 in `ACIP`: If a linear layer $l$ is not compressible for a given rate $r$ (i.e., a low-rank factorization would not save any parameters), we reset the corresponding mask parameters $\mathbf{p}_l$ to 1.0 and disable the low-rank adapter $\mathbf{\Delta}_l$. In this way, the layer is fully reset, and for $r = 1.0$, we exactly recover the base model. Note that this adapted rule is fully reversible and therefore still allows for Any Compression.

### D.4 Impact of the Stopping Criterion

In most experiments, we have used $r_{\text{stop}} = 0.4$ as maximum reasonable compression ratio, i.e., the pruning of masks is stopped if the size of the model is only 40% of the original one (measured in number of parameters of all target weight matrices). We have observed that at this point, the model performance has typically dropped so much that even a fine-tuned model would be of limited practical use.

Nevertheless, it is interesting to explore the sensitivity of compression-performance curves against different stopping ratios. The comparison shown in Figure A11 provides several insights in this respect: (i) "Forecasting" compressed models beyond the stopping ratio does not work very well, especially when stopping very

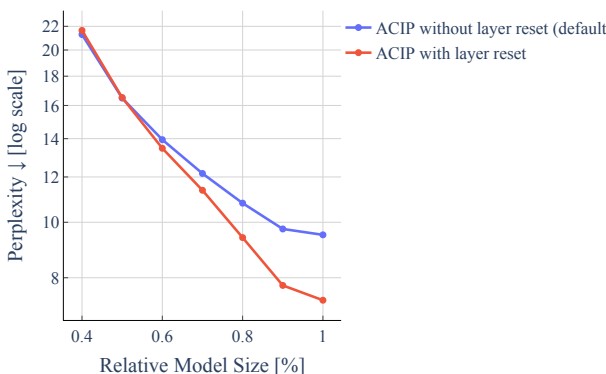

Figure A10: Compression-performance trade-off curves for **LLaMA-7B** on **C4** with and without resetting linear layers if they are incompressible for a given target compression rate $r$.

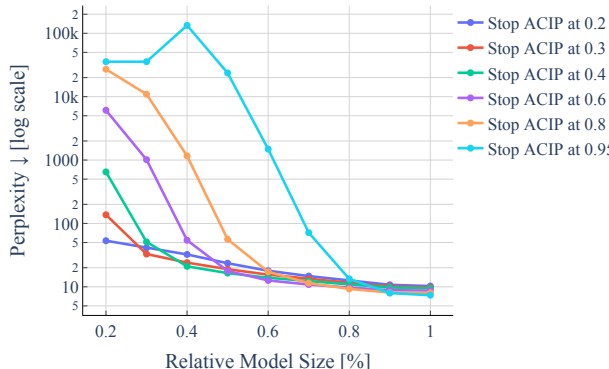

Figure A11: Compression-performance trade-off curves for **LLaMA-7B** on **C4**, using different stopping compression ratios $r_{\text{stop}}$ for ACIP.

early ($> 0.8$). (ii) The predictive capacity of ACIP remains valid for even stronger stopping compression ratios than 0.4. However, finding the largest reasonable stopping ratio is highly model-dependent. For less compressible models like LLaMA-3.1-8B, it could make sense to stop even earlier than 0.4 (cf. Figure 4). In general, we hypothesize that older models are more compressible than new ones, as the latter "carry" more information per weight due to significantly more training data (Allen-Zhu & Li, 2024).

### D.5 Impact of the Score Map – Forecasting Pruning Patterns

Here, we pick up the observation from Appendix D.4 that forecasting the performance of compressed models beyond the stop ratio leads to inaccurate predictions, i.e., the model is compressed more strongly than it has been done by ACIP itself. However, it turns out that the score map itself exhibits a certain forecasting capability. To this end, we run ACIP as usual until a stop ratio is reached, say $r_{\text{stop}} = 0.4$, but we stop updating the score map earlier in the optimization process. A few compression-performance curves with this modification are reported in Figure A12. We observe very similar curve shapes even if the score map is frozen after only a tiny fraction of mask parameters was pruned. This underpins our intuition from Section 2.2.2 that the pruning path of each parameter is fully determined at very early stage of ACIP.

### D.6 Impact of the Score Map – A Trivial One Does Not Work

There are certainly alternative ways to design useful score maps. For example, simply accumulating the gradients of all mask parameters entrywise over an ACIP-run works equally well as the strategy proposed in

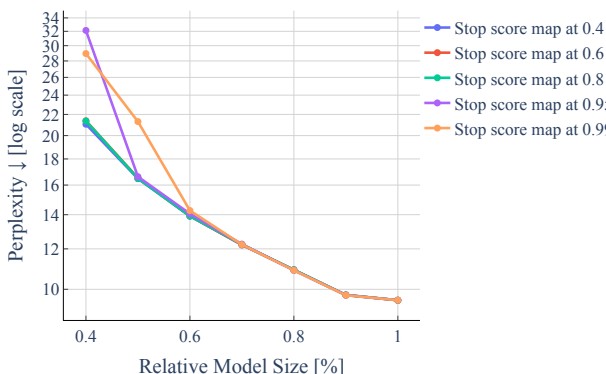

Figure A12: Compression-performance trade-off curves for **LLaMA-7B** on **C4**, stopping updates of the score map before the actual stopping criterion of `ACIP`.

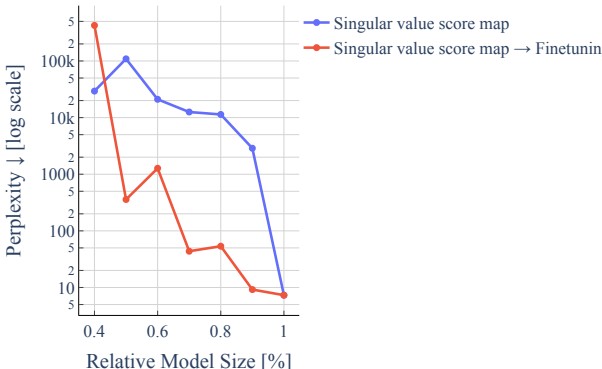

Figure A13: Compression-performance trade-off curves for **LLaMA-7B** on **C4**, using a trivial score map based on the initial singular values of the base model.

Section 2.2.2. It is therefore valid to ask whether one could even design score maps without any optimization. We demonstrate that perhaps the most obvious approach, namely setting the score map equal to the singular values of the weight matrices, does not work very well. Figure A13 shows that this training-free approach does not produce any reasonable compressed models and decent performance cannot be easily recovered with LoRA-finetuning. This simple experiment confirms that designing useful score maps is not a trivial endeavour and requires a carefully crafted algorithmic approach.

### D.7 Impact of Post-Tuning

Our main experiments are performed with 1k post-tuning steps in `ACIP` (see the description in Section 2.2.2 and Appendix B). Figure A14 shows analogous compression-performance trade-off curves for fewer or no post-tuning steps. We observe that post-tuning can indeed notably increase performance for higher compression ratios.

### D.8 Impact of Individual Layers – Example of LLaMA2-13B

As pointed out in the caption of Table A2, the linear layers targeted by `ACIP` were slightly modified for LLaMA2-13, namely all up projection layers were ignored. Figure A15 shows what would happen if they are compressed as well. While the performance predictions for $\geq 0.6$ look decent, the perplexity explodes for stronger compression; note that even additional fine-tuning does not recover a reasonable performance in this situation. We hypothesize that `ACIP` has pruned one or more singular values of the up projection layers

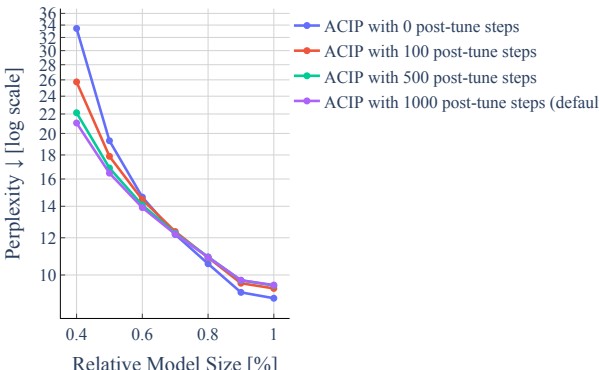

Figure A14: Compression-performance trade-off curves for **LLaMA-7B** on **C4** with different numbers of post-tuning steps in `ACIP`.

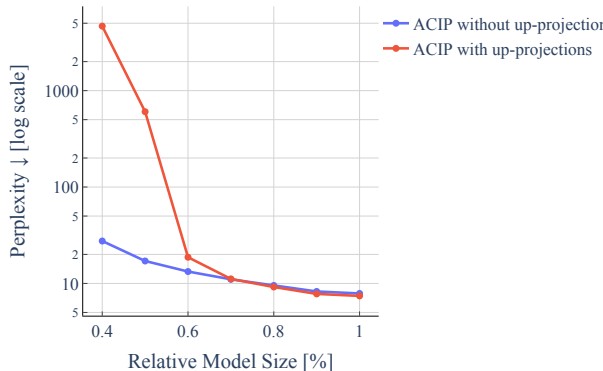

Figure A15: Compression-performance trade-off curves for **LLaMA2-13B** on **C4**, (not) ignoring the up projection layers in `ACIP`.

that are crucial for model's integrity. This finding might be related to the recent work by Yu et al. (2024) on pruning so-called super weights. In any case, `ACIP` is capable of revealing this undesirable behavior as demonstrated in Figure A15.

### D.9 Examples of Score Maps Generated by `ACIP`

Figure A16 and Figure A17 show two typical score maps generated by `ACIP` for LLaMA-7B and Qwen2.5-7B, respectively. A characteristic feature is that attention layers can be pruned more aggressively than the MLP layers. Similarly, we observe non-uniform pruning patterns for layers of the same type across all transformer layers. This confirms the findings of (Yuan et al., 2024; Jaiswal et al., 2024) and demonstrates that non-uniform structured compression can be achieved without any feature engineering.

### D.10 Examples of Generated Text by Compressed Models

Table A6 shows examples of generated text by compressed versions of LLaMA-7B.

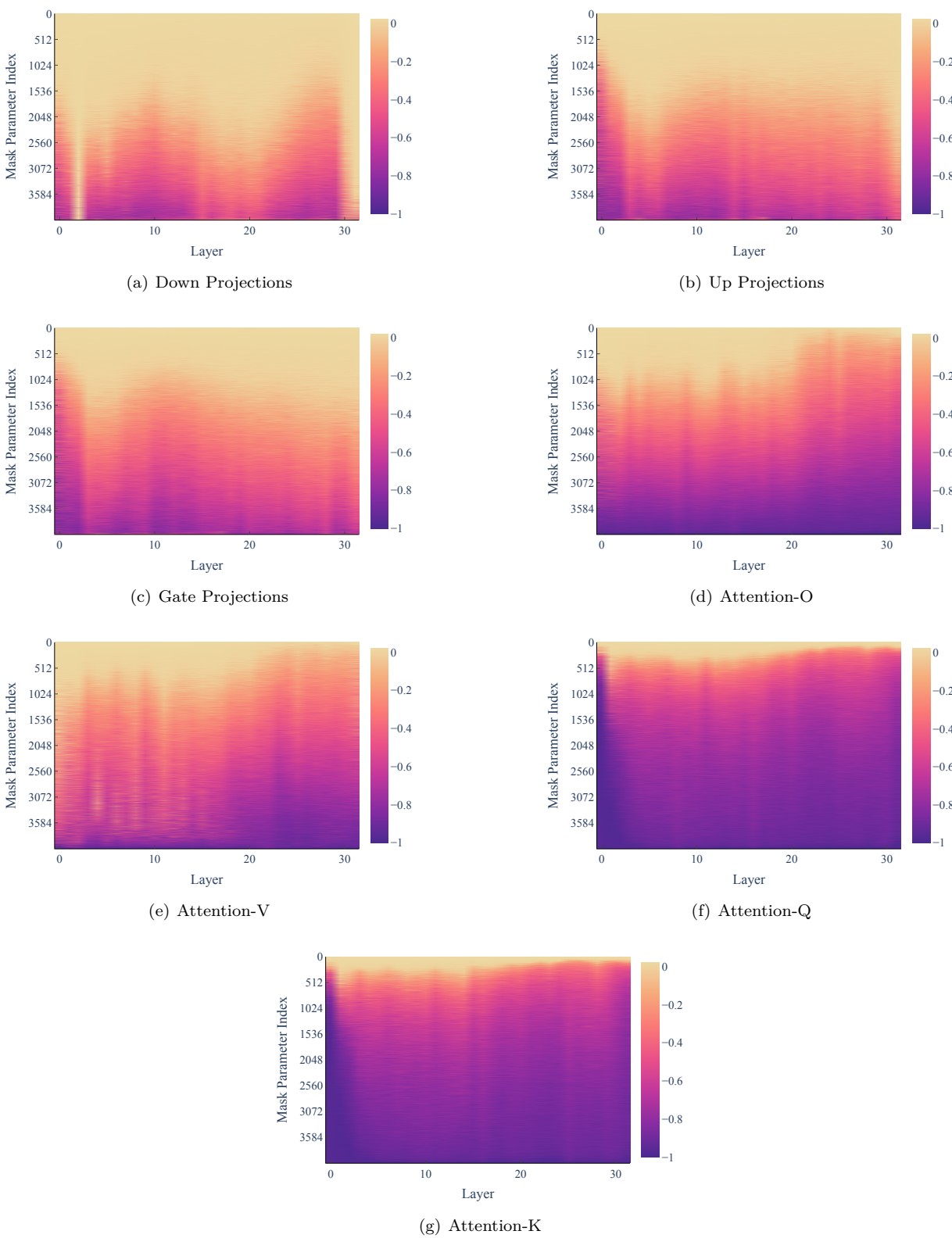

Figure A16: Example score maps generated by `ACIP` for **LLaMA-7B**. The negative values (cf. Algorithm 1) are normalized to −1 for the purpose of visualization.

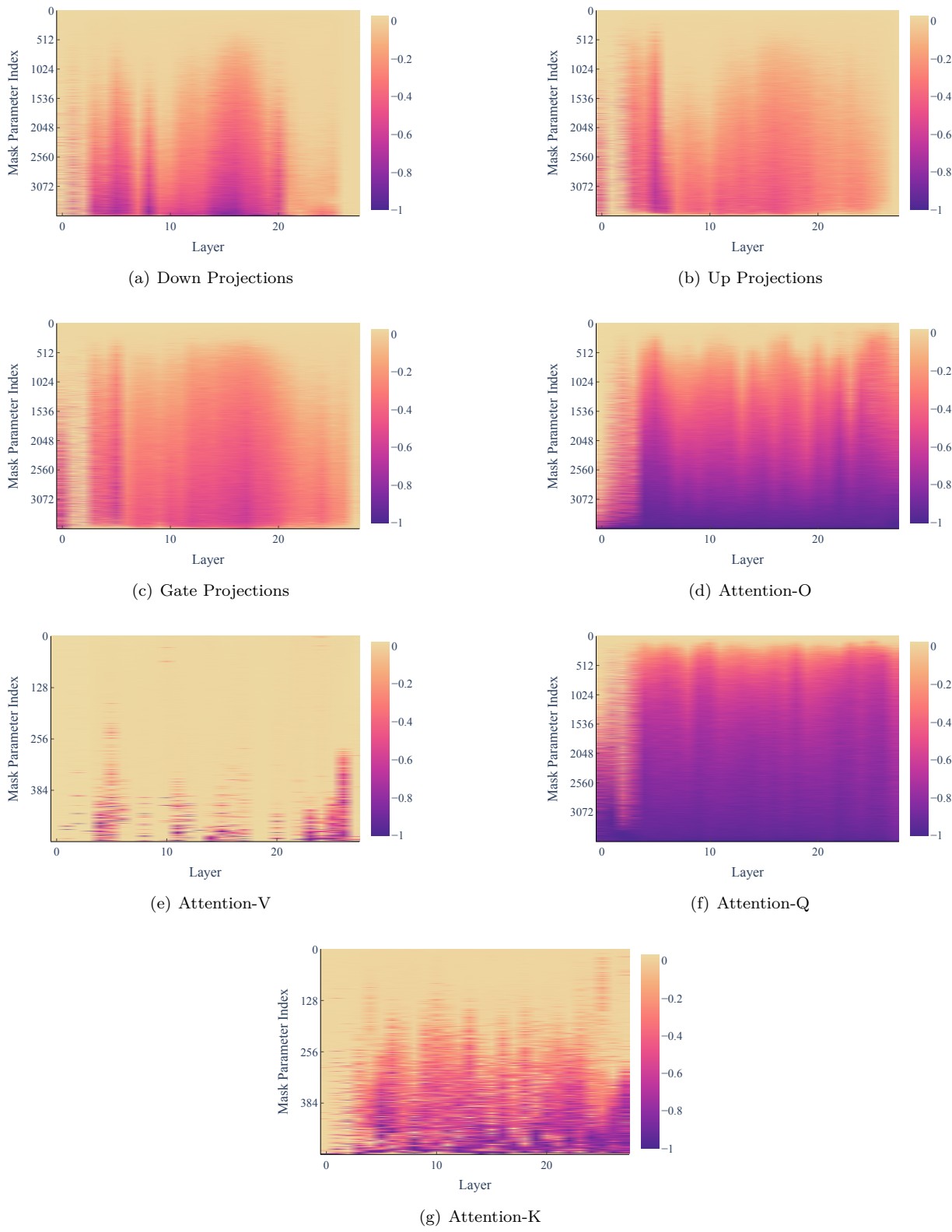

Figure A17: Example score maps generated by `ACIP` for **Qwen2.5-7B**. The negative values (cf. Algorithm 1) are normalized to −1 for the purpose of visualization.

Table A6: Example texts for two prompts generated for **LLaMA-7B** under different compressions produced by `ACIP`.

| Size | "How to compress Large Language Models?" | "What is the concept of time?" |
|---|---|---|
| Original | I am trying to run a large Language model generation, but I have a few files on disk of around 3gb each. I was wondering if there is a way to compress these files? I have searched the internet but have found nothing.
Re: How to compress Large Language Models?
I suggest compressing it to HDF5, you can find instructions on how to do this using TensorPacker and PNet on the TensorFlow Github repository . . . | Have you ever tried to ask a baby what it sees? Time, is it real? What is the concept of time? Have you ever tried to ask a baby what it sees? Let's see what they can understand.
The concept of time. What it is about, who discovered the idea of time, and how was it discovered? How is time measured? We can't define it. We can't point at what we are defining, we can't even show a picture of it . . . |
| 90% | Jeffrey Hunt, a linguist and applied deep learning researcher at the University of Technology, Sydney, talks with Caleigh Blackstock"s to discuss how to compress a large language model to speed up training and model execution.
What does the language model compression actually do to the overall model? . . . | Please let me know what is the concept of time.
The concept of time is all about the measurement of time. This measurement was first done using the hourglass.
When this hour Glass was invented, it was the only thing that could represent the passage of time. . . . |
| 80% | Jeffrey Shafman, Zsolt Nemeth, Jeffrey Shafman, Zsolt Nemeth, Jeffrey Shafman, Zsolt Nemeth. (2013). How to compress Large Language Models? In: ICML proceedings, Vol. 22, pp. 462-463.
Magnus Skallman. (2013). How to compress Large Language Models? In: ICML13 Proceedings, pp. 462-463. . . . | You can't even imagine that in that time,
The life of a tree in nature is as brief
As a flash of a lightning bolt,
But in this human world, it may endure
Through many millennia, if the will of man.
So tell me, what is the concept of time?
Hey, I've got one, and it's this: . . . |
| 70% | Apr 04, 2018 at 12:05 pm In the last 5-6 years, Natural Language Systems have been an important direction for my work and the research I do in the course of my PhD. I have been using them to teach myself Swedish and improve my Swedish by machine time in this post. Let's see if I've got it! . . . | Please tell me where is the concept of time in this poem. What is the significance of the poem "I Am Wine" by H.W.C. Carson"?
What does he mean by the poem title?
What does the stairs, the wine, & the man have in common?
What does he mean by "I am wine"?
What do you mean by "I am wine"? . . . |
| 60% | This is a 14 day free trial.
Please use the following link to sign in.
Select the country you need here to view pricing and purchase.
Whenever you use our web site, we will track your website traffic, the page where you first connected to our site, and the search keywords you use to find us. We'll use this information to help us continually improve our website. . . . | You may be wondering what is the concept of time.
The concept of time is all about the phenomenon of time. It says that there is never no time. It states that there is always a moment in time. At the moments of time is when we refer the time of life.
So, as a matter of the moment, time is just a point in space. . . . |
| 50% | Jeffrey Huan, Shu-Hua Ji, Xiao-Hong Li, Xiao-Xiao Li, Layin Ma"s. 2019:5, Springer.0000.0001/13/8/19/12/720-22-F43.
Miller A. "The Structure of the Language in Unsupervised Deep-Processing" in Journal of the Acoustical Actors (v. 11(6). 2006:, pp. 278-295).
Dong Ming, Sun Jin, He Mei-Chen,' . . . | Getting to this answer isn't that hard.
The concept of time is all about the study of time. The study of time isn't so hard to get a hold on.
When you're younger, you can begin thinking of how much you could have had if you had known how much time you had...if only we had it back then . . . |
| 40% | Apr 14, 2018 Resumes at The University of Florida will be released on Friday, May 29, 2018 2.05:53pm.
Cover your next job announcement with the help of our new Resume Template Builder.
To create your own Resume Template in minutes.
Improving the quality of your Resume.
To improve your Resume . . . | You may think that it is just a fancy word, or just the idea it had in the earlier world. But there exists a way to understand it.
To understand the idea of time by using the example of a clock, you can learn the very importance of time with a simple strategy.
The clock ticks with a watch. The clock has it time to operate. . . . |

