# OpenReview forum: "Choose Your Model Size: Any Compression of Large Language Models Without Re-Computation"
_TMLR — Accepted by TMLR_

### Review · Reviewer_h9RX · 2025-08-11

**Summary Of Contributions:**

The authors propose an algorithm ("Any Compression via Iterative Pruning" (ACIP)) for compressing networks. The algorithm works by computing a global score map upfront which then allows compression to any smaller size.
This is done by computing a score for the singular values of all linear layers, which then allows for choosing the top percentage to adjust compression ratio.
The authors provide an overview of the algorithm and validate it experimentally on real models with about 7B parameters (LLaMA, Qwen, Mistral)

**Audience:**

Yes

**Audience Explanation:**

This is a novel approach that is conceptually interesting and is also relevant to practical considerations.

**Broader Impact Concerns:**

No concerns.

**Claims And Evidence:**

Yes

**Claims Explanation:**

The writing and presentation are clear and the problem is well motivated.
The algorithm is validated on practically relevant models.

**Requested Changes:**

- If the authors have any insight on it, it would make sense to have a short discussion on how one wants to choose $\mathcal{L}$ in (8)
- There should be a more thorough discussion of the memory overhead and the corresponding cost of the algorithm compared to other algorithms. In particular it would be useful if it was clear how the cost (in terms of memory and compute) of ACIP in terms of model size compared to other algorithms.
- I am not quite convinced by A1 in appendix A, it is phrased as if the ACIP algorithm can lead to extra gains, however in Figure 6, 7 we never compare to a compression scheme without ACIP. If there is evidence that adding ACIP can improve another compression scheme in this context it would be good to add. This is particularly relevant for practical considerations.

---

> ### Author Response · Authors · 2025-08-27
> **Response to Reviewer h9RX**
>
> ### Role of the Loss Function
>
> > If the authors have any insight on it, it would make sense to have a short discussion on how one wants to choose L in (8)
>
> Thank you for this suggestion. We’ve added Remark 2.2 at the end of Section 2.2.2 to address the choice of loss function and calibration data. Please see there for more details.
>
> ### Memory Overhead
>
> > There should be a more thorough discussion of the memory overhead and the corresponding cost of the algorithm compared to other algorithms. In particular it would be useful if it was clear how the cost (in terms of memory and compute) of ACIP in terms of model size compared to other algorithms.
>
> We agree that these aspects are not discussed in sufficient detail in the original version of the paper. Following the reviewer’s suggestion, we have added a new subsection “Computational Considerations” (Section 2.3) that addresses these important points. Please see there for more details.
>
> ### Complementary to Other Compression Algorithms
>
> > I am not quite convinced by A1 in appendix A, it is phrased as if the ACIP algorithm can lead to extra gains, however in Figure 6, 7 we never compare to a compression scheme without ACIP. If there is evidence that adding ACIP can improve another compression scheme in this context it would be good to add. This is particularly relevant for practical considerations.
>
> We appreciate that the reviewer raises this concern, and we admit that the phrasing on “extra gains” in A1 is misleading. By “extra gain”, we were referring to our observation in Figure 6 that the compression capability of ACIP can be further improved by a quantization of the singular vector matrices. On the other hand, we do not claim that ACIP leads to extra gains on conventional quantization schemes and therefore we did not report any result in that direction. We revised the wording in A1 accordingly to avoid confusion about this aspect.

---

### Review · Reviewer_3m15 · 2025-08-19

**Summary Of Contributions:**

This paper proposes a new compression method for LLMs by leveraging a low-rank factorization strategy and decoupling pruning from the compression stage through a saved score map. The method aims to address two key challenges: (i) limitations caused by preset and discrete compression rates, and (ii) the need for repeated recomputation under different compression settings.

**Additional Comments:**

NA

**Audience:**

Yes

**Audience Explanation:**

1. The idea of pruning via singular values is novel and interesting.
2. The empirical results are strong and demonstrate the promise of the approach.

**Claims And Evidence:**

Yes

**Claims Explanation:**

1. The method is clearly described and easy to follow.
2. Experiments cover a wide range of model families to validate the pruning results.
3. The idea of decoupling score maps and reusing them in the compression stage is straightforward and effective.
4. However, the comparison with baseline methods is insufficient, and it is unclear why preset compression rates inherently cause the observed issues.
5. A concern is that the LoRA adapter may not yield sparsity in practice, as the forward pass could still involve dense matrix multiplications.
6. It would be helpful to understand the structure of the pruned matrix — whether the pruning pattern resembles structured pruning or unstructured pruning.

**Requested Changes:**

1. Provide a more thorough comparison with baseline methods, and clarify why preset compression rates are problematic.
2. Address the concern about LoRA adapters possibly remaining dense during forward computation.
3. Include wall-clock runtime or FLOPs as additional evaluation metrics, as they are critical for demonstrating practical efficiency.

---

> ### Author Response · Authors · 2025-08-27
> **Response to Reviewer 3m15**
>
> ### Baseline Methods and Preset Compression Rates
>
> > Provide a more thorough comparison with baseline methods, and clarify why preset compression rates are problematic.
>
> We agree that adding further baselines can underpin the competitiveness of our method (a related proposal was made by Reviewer hxCG as well). Following the reviewer’s suggestion, we have added two more approaches to the comparison in Table 1: Dobi-SVD as an even more recent work on  factorization-based LLM compression, and SVD-Magn., which builds a trivial score map based on singular value magnitudes. Compared to all other methods, SVD-Magn. allows for Any Compression analogously to ACIP and therefore serves as a natural baseline. The text of Section 3.2 was adapted accordingly.
>
> Clarification why preset compression rates are problematic: We thank the reviewer for this important point. The problem with methods tied to preset compression rates is the practical, computational barrier they create for users with specific deployment constraints. Real-world targets are often non-discrete (e.g., a maximum memory budget of 3.5 GB), but conventional methods are designed for discrete steps (e.g., 50% size or 4-bits). To meet a specific budget with such a tool, a user is forced into a costly "guess-and-check" cycle. They must repeatedly run the entire expensive compression algorithm with different preset rates until the resulting model happens to fit their precise constraint. This makes it computationally demanding or prohibitive to find the optimal model for a given budget.
> Our "Any Compression" approach aims to eliminate this barrier. After a single upfront computation, a model can be materialized to fit any specific budget in real-time, directly addressing the practitioner's need without repeated effort.
> To better emphasize this aspect in the manuscript, we have added the following sentence at the end of Problem 1 in the Introduction: “*In practice, preset compression rates are especially problematic when deploying under a specific, non-discrete constraint, e.g., a fixed memory budget, forcing users into a costly “guess-and-check” cycle.”*
>
> ### Structure of the Pruned Matrices
>
> > It would be helpful to understand the structure of the pruned matrix — whether the pruning pattern resembles structured pruning or unstructured pruning.
>
> Thanks for bringing this up. Compared to conventional unstructured pruning (e.g., dropping matrix entries) and structured pruning (e.g., dropping attention heads or matrix rows/columns), our low-rank decomposition approach plays a somewhat special role. Instead of representing a compressed linear layer as a single weight matrix, we use the singular-value decomposition as a (re-)parametrization. Since the SVD is a structured (unique) representation, we argue that pruning singular values falls into the category of structured pruning.
> So intuitively, it is most convenient to view the pruned matrices as a low-rank factorization into a “tall” and “short fat” matrix. This is analogous to the visualization that is commonly used for LoRA-type adapters.
>
> ### Concern About Merging LoRA
>
> > Address the concern about LoRA adapters possibly remaining dense during forward computation.
>
> This is a valid point that is not clarified in the original manuscript. While a naive merge of the pruned weight matrix with the LoRA adapters would indeed result in a dense matrix, one can avoid this sub-optimal representation by a concatenation that exploits the low-rankness of both components. For this, we have added Remark 2.3 at the end of Section 2.2.3. Please see there for more details.
>
> ### Reporting Efficiency Metrics
>
> > Include wall-clock runtime or FLOPs as additional evaluation metrics, as they are critical for demonstrating practical efficiency.
>
> We agree that these are very important metrics. Table A5 in Appendix D.1 provides insights in this regard: We report wallclock runtimes of each sub-step of the ACIP algorithm, as well as peak memory consumptions. Moreover, we have analyzed the inference efficiency of compressed models, measuring size, latency, and FLOPs.
>
> We further highlighted this part of the Appendix by an additional reference in Section 3.1 of the mainbody, and we have added a qualitative discussion of these aspects in the new Section 2.3.

---

### Review · Reviewer_hxCG · 2025-08-21

**Summary Of Contributions:**

The authors propose Any Compression via Iterative Pruning (ACIP), a post-training compression method utilizing singular value decomposition (SVD) to enable model size reduction without recomputation. ACIP decomposes linear layer weights via SVD, iteratively prunes singular values using a sparsity-inducing penalty, and creates a global score map to determine the pruning order for compression. Experiments verified ACIP's compression-model performance across three task types on multiple open-source models and its compatibility with quantization techniques.

Strength:
- The approach of generating a parameter score map to enable model compression to arbitrary target sizes without recomputation is novel.
- The experiments are comprehensive, spanning diverse datasets and models from three distinct LLM families at varying scales, while also integrating quantization techniques and fine-tuning.
- The manuscript is clearly written and well-organized.

Weakness:
- The definition of "feature importance" in Section 2.1.3 is ambiguous, as it does not clearly specify whether it refers to a parameter's contribution to loss optimization, sensitivity to model outputs, or another metric. A concise clarification of this term would improve reader understanding of the ACIP method's foundation.
- The hypothesis (Lines 207-210) is not sufficiently supported, as the parameter vanishing order may be influenced by confounding factors, undermining its reliability as an indicator of "importance." For instance, Equation (8) suggests that different initializations or $\lambda$ scale factors could lead to divergent optimization paths, resulting in inconsistent pruning sequences. Providing additional evidence or analysis to address these factors would bolster the hypothesis's credibility.
- Figure 3 lacks a legend, making the meaning of the colored lines unclear. Adding clear, descriptive labels would enhance readability and interpretation.
- The evaluation of state-of-the-art (SOTA) methods is limited, as it only compares two SVD-based baselines, potentially weakening the SOTA claim for factorization-based compression. Incorporating additional factorization techniques, such as tensor decomposition, would strengthen the evaluation's robustness and support broader conclusions.

**Additional Comments:**

Question:
- How sensitive is task performance to the choice of k (the number of pruned singular values)? The binary search ensures a target mathematical compression ratio but does not guarantee optimal model performance.
- How does the memory overhead of storing ACIP score maps scale with the number of model parameters? The substantial memory required to store scores for each parameter may limit ACIP's applicability to large-scale models, such as those with 70B or 670B parameters.

In addition, the paper can benefit from some discussion on the ethical implications of open-weight LLM compression.  For example, compressed models enable easier deployment on edge devices, lowering barriers for malicious use (e.g., real-time deepfakes, spam generation).

**Audience:**

Yes

**Audience Explanation:**

LLM compression methods like ACIP may be helpful to researchers and AI engineers who explored the feasibility of LLMs lightweights deployment.

**Claims And Evidence:**

Yes

**Claims Explanation:**

This paper proposes ACIP, an post-training compression method that enables arbitrary target compression rates through iterative singular value pruning. The method's advantages in preserving accuracy and generalization capability are both theoretically grounded in SVD principles and empirically validated through extensive experiments. However, the quality of the paper could be further enhanced by additional verification of both the hypothesis viability and the experimental comparison completeness.

**Requested Changes:**

1. Please address the concerns under the "Weaknesses" section.
2. Typos, for example, in  Line 197-198, the phrases (p = {...}, Δ = {...}, θ) lack verbs; in Line 201, missing preposition ("of") after "optimization".

---

> ### Author Response · Authors · 2025-08-27
> **Response to Reviewer hxCG**
>
> ### Feature Importance
>
> > The definition of "feature importance" in Section 2.1.3 is ambiguous, as it does not clearly specify whether it refers to a parameter's contribution to loss optimization, sensitivity to model outputs, or another metric. A concise clarification of this term would improve reader understanding of the ACIP method's foundation.
>
> Thanks for bringing up this point, as it might not be clear to readers who are not familiar with l1-regularization. We added the following explanation of the term “feature importance” to Section 2.1.3: *Here, \`feature importance' is understood as the relative contribution to the (task- and data-specific) loss $\\mathcal{L}\\bigl(X; \\boldsymbol\\theta\\bigr)$, i.e., a feature is considered more important if its removal causes a larger increase in the loss. In particular, different choices of $\\mathcal{L}$ may lead to different solution paths and feature rankings.*
>
> ### Robustness of Score Maps
>
> > The hypothesis (Lines 207-210) is not sufficiently supported, as the parameter vanishing order may be influenced by confounding factors, undermining its reliability as an indicator of "importance." For instance, Equation (8) suggests that different initializations or  scale factors could lead to divergent optimization paths, resulting in inconsistent pruning sequences. Providing additional evidence or analysis to address these factors would bolster the hypothesis's credibility.
>
> Thank you for this comment. We agree that a certain robustness across various conditions is crucial.
> Considering the robustness of the score map, we find it less important than overall task importance. In fact, we consider the score map to be a useful tool for compression, however, the exact instantiation of the score map is less important as long as downstream performance is robust. We therefore focused our experiments on the analysis and evaluation of the compression-performance trade-off. While we anticipate minor variations in score maps due to different random seeds, we expect this to have a negligible impact on the compression-performance trade-off. To substantiate this, we will run ACIP on Llama-7B with five distinct random seeds, plot the compression-performance trade-off for each run, and include these findings in the final version of the paper.
> Apart from that, we acknowledge the sensitivity of the scale factor's initialization, which led to our proposal of a simple scheduler (see Remark 2.1) that consistently performs well with a fixed set of hyperparameters across all models. We have observed that ACIP maintains robustness under the same hyper-parameter conditions for all models, as illustrated in Figures 4 and 5\.
>
> ### Legend of Figure 3
>
> > Figure 3 lacks a legend, making the meaning of the colored lines unclear. Adding clear, descriptive labels would enhance readability and interpretation.
>
> Good catch\! Each line corresponds to a different feature, so the color coding is not important at all. To avoid any confusion, we removed the color coding and added a clarifying sentence to the caption: *“Each line corresponds to the evolution of a parameter value over training time.”*
>
> ### Evaluation of SOTA
>
> > The evaluation of state-of-the-art (SOTA) methods is limited, as it only compares two SVD-based baselines, potentially weakening the SOTA claim for factorization-based compression. Incorporating additional factorization techniques, such as tensor decomposition, would strengthen the evaluation's robustness and support broader conclusions.
>
> Thank you for this suggestion. We surveyed the literature for recent non-SVD-based work in factorization-based compression, in particular, tensor decomposition methods that have been applied to LLMs. We found that existing works are evaluated on different model architectures and/or benchmarks, and often do not have public codebases, for example, see [https://arxiv.org/abs/2501.19135](https://arxiv.org/abs/2501.19135), [https://arxiv.org/abs/2405.06626](https://arxiv.org/abs/2405.06626), [https://arxiv.org/abs/2501.15674](https://arxiv.org/abs/2501.15674), [https://arxiv.org/abs/2307.00526](https://arxiv.org/abs/2307.00526). A fair and rigorous comparison (in the context of Table 1\) would therefore require a full re-implementation of each method, which we believe is beyond the scope of this rebuttal.
> Nevertheless, we agree that adding further baselines can strengthen the SOTA claims of our work (a similar comment was made by Reviewer 3m15, too). For this reason, we have added two more approaches to the comparison in Table 1: Dobi-SVD as an even more recent work on factorization-based LLM compression, and SVD-Magn., which builds a trivial score map based on singular value magnitudes. Compared to all other methods, SVD-Magn. allows for Any Compression analogously to ACIP and therefore serves as a natural baseline. The text of Section 3.2 was adapted accordingly.
>
> Please find the second part of our response below.

---

> > ### Author Response · Authors · 2025-08-27
> > **Response to Reviewer hxCG (continued)**
> >
> > ### Typos
> >
> > > Typos, for example, in Line 197-198, the phrases (p \= {...}, Δ \= {...}, θ) lack verbs; in Line 201, missing preposition ("of") after "optimization".
> >
> > Thanks, we fixed them.
> >
> > ### Additional Comment: Choice of k
> >
> > > How sensitive is task performance to the choice of k (the number of pruned singular values)? The binary search ensures a target mathematical compression ratio but does not guarantee optimal model performance.
> >
> > Our experiments in Section 3 show a monotonic relationship between the compression rate (model size) and task performance. This monotonicity also extends to the trade-off between $k$ and task performance, as there is a monotonic functional relationship between $k$ and the compression rate $r$ (see Section 2.2.3). Consequently, the sensitivity of downstream performance to $k$ is implicitly captured by the compression-performance trade-offs detailed in Section 3.1.
> > It is important to note that $k$ represents the total number of pruned singular values across all parameterized linear layers. The number of pruned singular values can vary significantly between layers (see Figures A16 & A17). Therefore, we consistently report the compression rate for its intuitive and easily interpretable nature.
> >
> > ### Additional Comment: Efficient Storage of Score Maps
> >
> > > How does the memory overhead of storing ACIP score maps scale with the number of model parameters? The substantial memory required to store scores for each parameter may limit ACIP's applicability to large-scale models, such as those with 70B or 670B parameters.
> >
> > This is a very valid question to ask, but fortunately no strong concern for our approach. Indeed, the total number of singular values (and so the size of the score map) only scales linearly with the feature dimension of layers and model depth (see also Line 184-186 in the original manuscript). More specifically, the size of the score maps can be computed by $\\sum\_l (\\min(m\_l, n\_l))$, where $l$ is the index over all weight matrices $W\_l$,  and $m\_l$ and $n\_l$ are the number of rows and columns of $W\_l$, respectively.
> > For instance, a LLaMA-7B model's score map contains approximately 900K parameters. Due to the linear scaling, the size of a score map for a 670B model would be in the range of about 100M parameters (exact numbers depend on the specific architecture), which is tractable.
> >
> > ### Additional Comment: Ethical Implications
> >
> > > In addition, the paper can benefit from some discussion on the ethical implications of open-weight LLM compression. For example, compressed models enable easier deployment on edge devices, lowering barriers for malicious use (e.g., real-time deepfakes, spam generation).
> >
> > We appreciate this suggestion and have incorporated a "Broader Impact Statement" section at the end of the manuscript.

---

### Author Response · Authors · 2025-08-27
**Response to Reviewers**

We would like to sincerely thank all reviewers for their reports and positive feedback! All comments and suggestions have helped to improve the presentation and clarity of the paper. We hope that all raised concerns of the reviewers are satisfactorily addressed by our individual responses. We have uploaded a revised version of the manuscript with corresponding changes highlighted in blue.

---

### Decision · Action_Editor_e1M5 · 2025-10-09

**Recommendation:** Accept as is

**Audience:**

Yes

**Audience Explanation:**

The paper introduces a compression approach for LLMs, a topic of broad relevance to the ML community.

**Claims And Evidence:**

Yes

**Claims Explanation:**

This paper presents a compression method for LLMs that uses low-rank factorization and a saved score map to decouple pruning from compression. The method addresses two core challenges: constraints from predefined compression rates and the inefficiency of re-computation across varying settings.

While reviewers pointed out the lack of additional comparisons with other methods to better assess the approach’s utility, and the authors partially addressed this during the rebuttal phase, all reviewers agree the proposed method is novel and supported by enough clear and convincing experimental evidence.

Due to all of the above, I recommend acceptance.